# Effective in vivo binding energy landscape illustrates kinetic stability of RBPJ-DNA binding

Duyen Huynh[1,7], Philipp Hoffmeister[2,7], Tobias Friedrich[3,4,5], Kefan Zhang[1], Marek Bartkuhn[4,5], Francesca Ferrante[3], Benedetto Daniele Giaimo[3], Rhett A. Kovall[6], Tilman Borggrefe[3], Franz Oswald[2]✉ & J. Christof M. Gebhardt[1]✉

Transcription factors (TFs) such as RBPJ in Notch signaling bind to specific DNA sequences to regulate transcription. How TF-DNA binding kinetics and cofactor interactions modulate gene regulation is mostly unknown. We determine the binding kinetics, transcriptional activity, and genome-wide chromatin occupation of RBPJ and mutant variants by live-cell single-molecule tracking, reporter assays, and ChIP-Seq. Importantly, the search time of RBPJ exceeds its residence time, indicating kinetic rather than thermodynamic binding stability. Impaired RBPJ-DNA binding as in Adams-Oliver-Syndrome affect both target site association and dissociation, while impaired cofactor binding mainly alters association and unspecific binding. Moreover, our data point to the possibility that cofactor binding contributes to target site specificity. Findings for other TFs comparable to RBPJ indicate that kinetic rather than thermodynamic DNA binding stability might prevail in vivo. We propose an effective in vivo binding energy landscape of TF-DNA interactions as instructive visualization of binding kinetics and mutation-induced changes.

Transcription factors (TFs) such as the DNA-binding hub RBPJ in Notch signal transduction are vital for the regulation of gene expression as they identify specific DNA target sequences and trigger events culminating in gene repression or activation. The efficiency of gene regulation depends among others on the time the TF needs to find its target sequence (target site search time)[1,2] and the time it stays bound to the target sequence (residence time)[1–8]. To which of its potential target sites a TF associates relies on various traits including the accessibility of the site and the presence of other TFs or cofactors[9–11], wherefore TFs only occupy a subset of their target sequences[12,13]. Mutations of the TF affecting its activation or repression capability may give rise to altered DNA binding kinetics. While a mutation within

the DNA binding interface is expected to alter the residence time of a TF, it has been observed that such mutations are able to affect the target site search time of a TF[1,2,14,15]. Therefore, predicting the mechanism of action of a mutated TF from the position of the mutation within the domain-structure of the TF is challenging.

Notch signaling is a short-range cell-to-cell communication pathway in metazoan species evolutionarily conserved from *C. elegans* to *H. sapiens*[16]. It plays a pivotal role in both embryonic development and adult tissue homeostasis[17]. Transcription regulation in the Notch signaling pathway is mediated by the DNA binding factor RBPJ (recombination signal binding protein for immunoglobulin kappa J region), which functions as the core DNA binding hub[18]. Its DNA binding interface

¹Institute of Experimental Physics and IQST, Ulm University, Ulm, Germany. ²Clinic of Internal Medicine I, University Medical Center Ulm, Ulm, Germany. ³Institute of Biochemistry, Justus-Liebig-Universität Gießen, Gießen, Germany. ⁴Biomedical Informatics and Systems Medicine, Justus-Liebig-Universität Gießen, Gießen, Germany. ⁵Institute for Lung Health (ILH), Gießen, Germany. ⁶Department of Molecular Genetics, Biochemistry and Microbiology, University of Cincinnati College of Medicine, Cincinnati, OH, USA. ⁷These authors contributed equally: Duyen Huynh, Philipp Hoffmeister. ✉e-mail: franz.oswald@uni-ulm.de; christof.gebhardt@uni-ulm.de

comprises an N-terminal and a beta-trefoil (BTD) domain (Supplementary Fig. 1). The BTD and the C-terminal domain are involved in the interaction with cofactors[19]. In the absence of Notch signaling, RBPJ functions as a repressor of Notch target genes by recruiting specific corepressor components such as SHARP (SMRT/HDAC1-associated repressor protein)[20–22] to form a transcription repressor complex. With active Notch signaling, the Notch Intracellular Domain (NICD1, hereinafter referred to as NICD) is released from the membrane by γ-secretase cleavage and translocates into the nucleus[23], where it forms a transcription activation complex with the cofactor MAML[24,25]. The cofactor composition therefore determines whether the complex around DNA-bound RBPJ acts as a repressor or activator of Notch signaling.

A deregulated Notch signaling pathway is responsible for severe congenital diseases such as Alagille Syndrome[26] or Adams-Olivier-Syndrome (AOS)[27]. AOS is associated with the lysine to glutamic acid mutation K195E in the DNA binding interface of RBPJ[27,28]. Numerous questions related to the signal transfer mechanism of Notch signaling remain. For example, it is unclear whether RBPJ is sufficient to identify specific target sites on chromatin, or whether cofactors are involved in target site selection. Further, it is unknown how cofactors or mutations

such as K195E contribute to or alter the DNA binding characteristics of RBPJ in vivo.

Here, we determine at repressive conditions the target site search time, the DNA residence time, and the ability for functional binding of RBPJ and of several DNA and cofactor binding mutant variants by live-cell single-molecule tracking and in vivo reporter assays, and measured their genome-wide chromatin occupation by ChIP-Seq. Our measurements revealed important insight into the DNA binding kinetics and specificity of RBPJ and other TFs.

## Results

### Both DNA and cofactor binding contribute to the functional binding of RBPJ

To characterize cofactor-dependent DNA binding of RBPJ, we introduced various mutations into the DNA or cofactor binding interfaces. To disturb DNA binding, we chose the mutation R218H (R/H)[29], the DNA binding mutation K195E (K/E) found in patients with AOS[28], or the triple mutation K195E/R218H/S221D (KRS/EHD) combining the phosphomimetic S221D[30] with R/H and K/E[28] (Fig. 1a and Supplementary Fig. 1). To disturb cofactor binding, we chose the double mutation

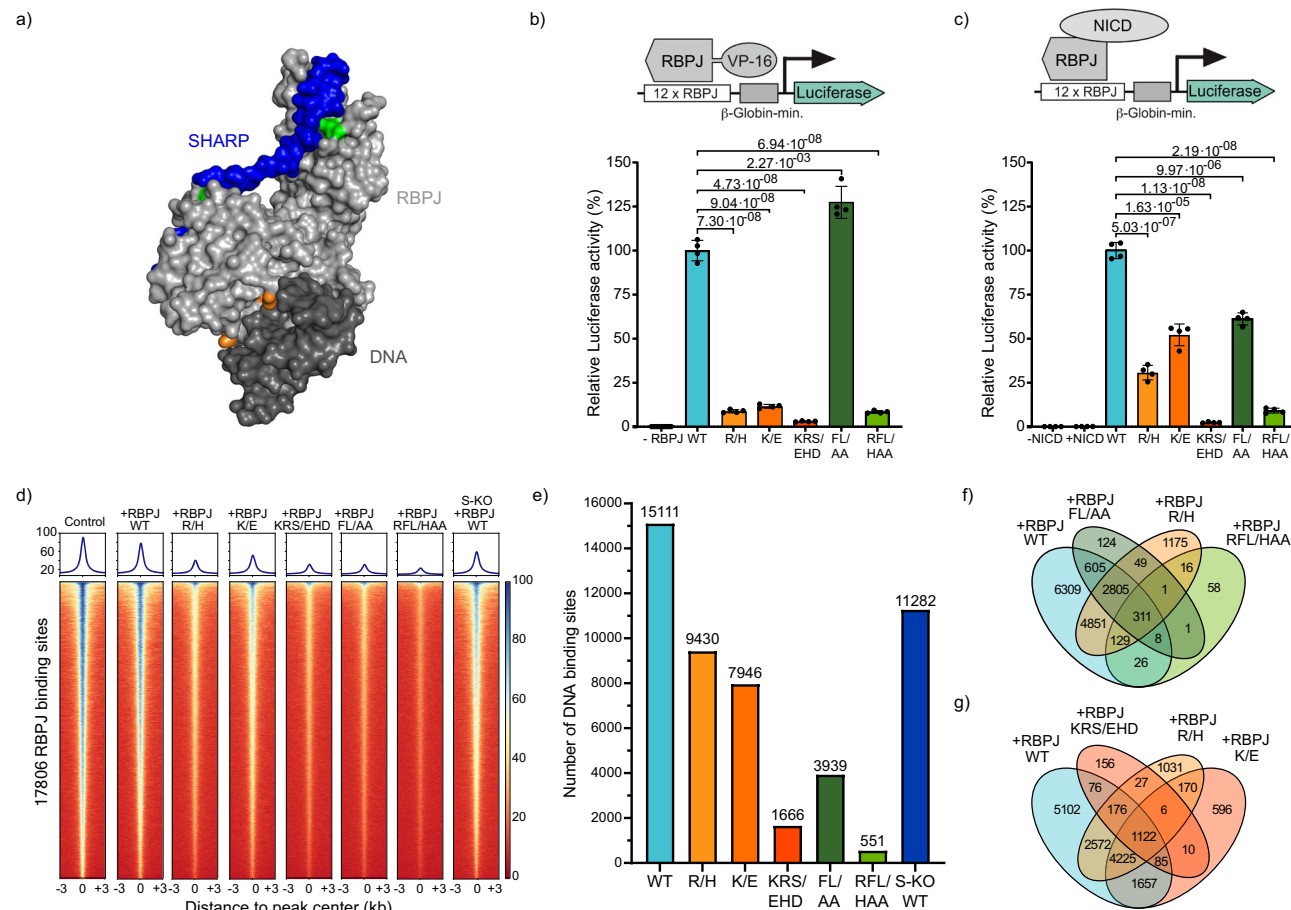

**Fig. 1 | Interactions of RBPJ with cofactors and DNA determine transactivation activity and binding specificity. a** Surface representation of a RBPJ-SHARP complex structure (light gray/blue) bound to DNA (dark gray) (PDB entry: 6DKS). Amino acid mutations in interfaces of cofactor binding (green) and DNA-binding (orange) are highlighted. In vivo luciferase activity assays. Relative luciferase activity of RBPJ-specific reporter constructs in RBPJ knockout cell line transfected with **b** RBPJ-VP16 fusion variants or **c** RBPJ variants and co-transfected with NICD (mean values ± s.d.; $n = 4$ independent experiments). *P*-values were determined using two-tailed, unpaired Student's *t*-test. Insets: Scheme of the reporter constructs and activating moieties. For details of variants see explanation in the text. -RBPJ: control with untransfected RBPJ knockout cell line, −/+NICD: control with untransfected/NICD-

transfected RBPJ knockout cell line. **d** Heatmap of ChIP-seq reads from RBPJ knockout cell line transduced with HT-RBPJ variants with reads centered around RBPJ binding sites called in an untransduced HeLa cell line control. ChIP was performed with an antibody against RBPJ. Read number is color-coded. For details of variants see the explanation in the text. S-KO: ChIP-seq reads from SHARP knockout cell line transfected with HT-RBPJ. **e** Number of binding sites called for HT-RBPJ variants in RBPJ knockout cell line or SHARP knockout cell line. **f, g** Venn diagrams depicting common binding sites of indicated HT-RBPJ variants. Color code for RBPJ variants in **b, c, e–g**: RBPJ WT: light blue, R/H: yellow, K/E: orange, KRS/EHD: red, FL/AA: dark green, RFL/HAA: light green, S-KO RBPJ-WT: dark blue. Source Data are provided as a Source Data file for Fig. 1b, c, e.

F261A/L388A (FL/AA)[22], which affects SHARP binding, and the triple mutation R218H/F261A/L388A (RFL/HAA), which disrupts both, DNA- and cofactor binding. Since SHARP is an important cofactor within the repressor complex of RBPJ, we also assessed how the absence of SHARP influenced RBPJ binding.

We first tested the ability of RBPJ variants to functionally bind to the canonical RBPJ target sequence a/GTGGGAAa at repressive consitions[31]. Therefore, we fused the RBPJ variants to the transcription activation domain VP16[32] and utilized a luciferase-based transcription reporter assay in an RBPJ-depleted HeLa cell line, clone #42[33] (Fig. 1b and Methods). Importantly, this common approach[34,35] tests the functionality of binding of RBPJ variants even in the absence of NICD, as the activation domain VP16 recruits the transcriptional machinery if the fusion protein is bound to DNA. As expected, the mutations in the DNA binding interface reduced the luciferase signal to either low (R/H, K/E, and RFL/HAA) or very low (KRS/EHD) values above the back-ground (Fig. 1b). In contrast, solely disturbing the cofactor binding interface (FL/AA) increased the transcription activity, presumably due to impeded binding of corepressor SHARP. This suggests that an intact DNA binding interface is required for functional binding of RBPJ to its canonical site.

To further assess the role of cofactor binding on the transcription activity of RBPJ in activating conditions, we repeated our luciferase assay in the presence of NICD as coactivator, instead of VP16 (Fig. 1c and Methods). As expected, mutation FL/AA reduced the transcriptional activity compared to RBPJ-WT. Surprisingly, low transcription activation in the presence of the DNA binding mutations R/H and K/E could be partially rescued by NICD. This was not possible for the triple mutation KRS/EHD or the mutation RFL/HAA that disturbs both DNA and cofactor binding. Thus, the cofactor NICD activates transcription more than VP-16 in this context, if DNA binding is only slightly disturbed.

## Cofactor binding might play a role in the target site specificity of RBPJ

Next, we tested the chromatin binding of RBPJ variants at repressive conditions on a genome-wide scale. For later visualization in live cells, we fused wild-type RBPJ (RBPJ-WT) and the RBPJ mutants to an N-terminal HaloTag (HT)[36]. We confirmed binding of HT-RBPJ-WT and HT-RBPJ-KRS/EHD to NICD and SHARP and transcription activity of HT-RBPJ-WT (Supplementary Fig. 2). As expected, HT-RBPJ variants with disturbed cofactor binding interface (FL/AA and RFL/HAA) showed impeded binding to NICD and SHARP (Supplementary Fig. 2). We introduced HT-RBPJ variants into the RBPJ-depleted HeLa cell line clone #42 by lentiviral transduction (Supplementary Fig. 3a and Methods). Expression levels varied between 1.4 and 1.8-fold of endo-genous RBPJ in HeLa cells (Supplementary Fig. 3b, c). Of note, while RBPJ variants with disturbed DNA binding interface showed pre-dominant nuclear localization, as expected, variants with disturbed cofactor binding interface additionally localized to the cytoplasm (Supplementary Fig. 3a). We further introduced HT-RBPJ-WT into a SHARP-depleted cell line (clone #30, Supplementary Fig. 4). ChIP-Seq reproduced the core binding motif of RBPJ in control cells and con-firmed the absence of RBPJ binding in RBPJ-depleted cells (Supple-mentary Fig. 5a-c). We further revealed the binding profile of HT-RBPJ variants in RBPJ-depleted cell lines for RBPJ target genes (Supple-mentary Fig. 5d) and confirmed normal RBPJ binding in SHARP-depleted cells (Supplementary Fig. 5e-f).

HT-RBPJ-WT closely reproduced the binding profile of endogen-ous RBPJ (Fig. 1d). In accordance with compromised DNA binding, the DNA binding mutations R/H, K/E and KRS/EHD reduced the DNA binding of RBPJ genome-wide (Fig. 1d). Moreover, they reduced the number of identified binding sites of HT-RBPJ-WT by ~38%, ~47%, and ~89%, respectively (Fig. 1e). Surprisingly, the mutations FL/AA in the cofactor binding interface also reduced the intensity of RBPJ binding

(Fig. 1d) and reduced the number of binding sites by ~74% (Fig. 1e). Accordingly, for the triple mutation RFL/HAA, almost no binding sites were called. However, we cannot exclude that reduced nuclear levels of RBPJ variants FL/AA and RFL/HAA affected the ChIP-Seq results. In the absence of SHARP the number of RBPJ sites identified was also reduced. Overexpression of RBPJ variants compared to endogenous RBPJ might artificially increase the number of identified binding sites. Thus, the values we obtained represent an upper limit. However, the number of binding sites identified in the endogenous RBPJ control (17,806) was larger than the one in the HT-RBPJ-WT reconstitution (15,111), indicating that the number of identified binding sites was not overestimated at our level of overexpression. The position of observed sites in all RBPJ mutants largely overlapped with those of HT-RBPJ-WT, with less than ~15% de novo binding to off-target sites (Fig. 1f, g and Supplementary Fig. 6). Taken together, our ChIP-Seq data point to the possibility that not only DNA binding but also cofactor binding of RBPJ contributes to specifying chromatin binding of this transcription fac-tor to a fraction of its target sites.

## The DNA residence time decreases for DBD mutants of RBPJ
Next, we quantified the dissociation rate of HT-RBPJ-WT from chro-matin at repressive conditions by live-cell single-molecule tracking in the RBPJ-depleted HeLa cell line clone #42 (Fig. 2b). We visualized individual HT-RBPJ-WT molecules by covalently labeling the HaloTag with HT-SiR dye[37], excited fluorescence using HILO illumination to optimize the signal-to-noise ratio[38], and detected and tracked fluor-escent molecules using the program TrackIt[39]. We selected cells for imaging that exhibited comparable molecule counts (Supplemen-tary Fig. 7).

To measure the dissociation rate, we applied several repetitive time-lapse illumination schemes, each comprising an image of 50 ms camera integration time and a time-lapse-specific frame cycle time between 0.1 s and 6.4 s[33] (Fig. 2a). This approach allows correcting for tracking errors and photobleaching of fluorophores, and ensures covering a broad temporal bandwidth[39]. We identified binding events as prolonged persistence of fluorescently labeled molecules (Fig. 2b, Supplementary Movie 1, and Methods) and collected the fluorescence survival times in histograms (Fig. 2c). By performing an inverse Laplace transformation using the GRID method[40], we extracted both the event and the state spectrum (Fig. 2d) of dissociation rates from these sur-vival time distributions. The event spectrum informs on the number of bound molecules per time showing a certain dissociation rate, while the state spectrum informs on the fraction of bound molecules within a snapshot of time. The dissociation rate spectrum of HT-RBPJ-WT consisted of several distinct rate clusters, corresponding to different residence time classes on chromatin (Fig. 2d).

To gain more information on the possible origin of the dissocia-tion rate clusters of HT-RBP-WT, we also measured the dissociation rate spectrum of HT-RBPJ-KRS/EHD (Fig. 2c, d). This RBPJ variant showed minimal functional binding and transcriptional activity in luciferase assays (Fig. 1b, c) and showed strongly reduced binding to RBPJ loci in ChIP-Seq, without de novo binding (Fig. 1d, e, g). It therefore barely binds specifically to DNA. For HT-RBPJ-KRS, the very long binding events present in HT-RBPJ-WT were missing (Fig. 2c, d). We therefore reasoned that the longest residence time class of HT-RBPJ-KRS/EHD and the corresponding dissociation rate cluster might mark the border between unspecific binding of the DNA binding mutant and the specific binding regime of RBPJ. We thus used the dissociation rate spectrum of HT-RBPJ-KRS/EHD to sort the dissocia-tion rates of HT-RBPJ variants into dissociation from either unspecific or specific binding sites (Fig. 2d). However, we cannot exclude that short binding events of RBPJ-WT also have functional consequences. We then calculated the average specific and unspecific residence times for HT-RBPJ-WT and -KRS/EHD from the inverse of the weighted sum of the corresponding dissociation rates. For HT-RBPJ-WT, we determined

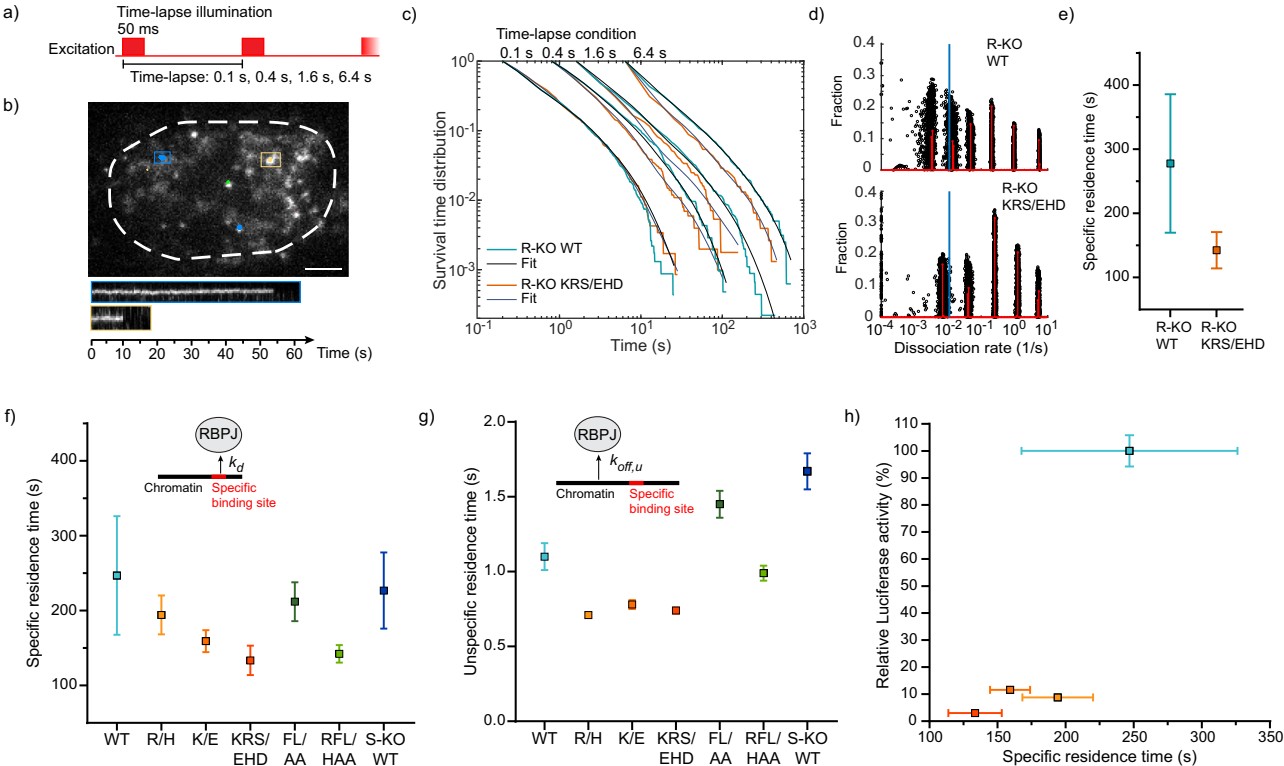

**Fig. 2 | Residence times of HT-RBPJ variants. a** Scheme of illumination patterns in time-lapse measurements with indicated camera integration and frame cycle times. **b** Tracks of single HT-RBPJ-WT molecules overlaid with an example image of a 100 ms time-lapse movie (Supplementary Movie 1) and kymographs of indicated molecules. Scale bar is 4 μm. **c** Survival time distributions of HT-RBPJ-WT (light blue lines) and HT-RBPJ-KRS/EHD mutant (orange lines) at time-lapse conditions shown on top and survival time function obtained with GRID (black lines). For experimental statistics see Supplementary Table 3. **d** State spectra of dissociation rates of HT-RBPJ-WT and HT-RBPJ-KRS/EHD obtained with GRID using all data (red). As an error estimation GRID was run 500 times with each run using 80 % of the data (black circles). The blue line indicates the dissociation rate of 0.01 s⁻¹. For experimental statistics see Supplementary Table 3. **e** Specific residence times of HT-RBPJ WT and the DNA triple mutant KRS/EHD in Hela RBPJ knock-out cells (inverse of $k_d$, the

weighted average of dissociation rates in **d**) below 0.01 s⁻¹). Error bars denote s.d. from the resampled spectra in **d**). Statistics are provided in Supplementary Table 3. **f** Specific residence times of HT-RBPJ variants (inverse of $k_d$, the weighted average of dissociation rates below 0.01 s⁻¹) (Supplementary Table 2). Error bars denote s.d. of the resampled data. Inset: sketch of RBPJ dissociation from a specific binding site with rate constant $k_d$. Statistics are provided in Supplementary Table 3. **g** Unspecific residence times of HT-RBPJ variants (inverse of $k_{off,u}$, the weighted average of dissociation rates above 0.01 s⁻¹) (Supplementary Table 2). Error bars denote s.d. of the resampled data. Inset: sketch of RBPJ dissociation from an unspecific site with rate constant $k_{off,u}$. Statistics are provided in Supplementary Table 3. **h** Relative luciferase activity of HT-RBPJ-VP16 fusion variants versus their specific residence time (replot of data from Figs. 1b and 2f). Color code in **f, g, h** as in Fig. 1. Source Data are provided as a Source Data file for Fig. 2c–h.

unspecific ($\tau_u$) and specific ($\tau_s$) residence times of $(0.9 \pm 0.1)$ s and $(277 \pm 108)$ s, respectively (Supplementary Tables 1–3).

We next characterized to which extent the various mutations in the DNA or cofactor binding interface altered the dissociation of RBPJ from chromatin. To be able to compare the results with those in a SHARP-depleted cell line that still expressed endogenous RBPJ, we stably inserted the HT-RBPJ variants into HeLa cells containing endogenous RBPJ. All tagged variants with disturbed DNA binding localized to the nucleus as expected from endogenous RBPJ, while the variants with disturbed cofactor binding interface again showed additional cytoplasmic distribution (Supplementary Fig. 8a). All variants showed comparable overexpression between 92% and 130% of the expression level of endogenous RBPJ (Supplementary Fig. 8b–d). Control measurements revealed comparable residence times of HT-RBPJ-WT or HT-RBPJ-KRS/EHD in the presence and absence of endogenous RBPJ (Fig. 2e, f).

As expected from compromised DNA binding, the mutations in the DNA binding interface reduced the specific residence times $\tau_s$ on chromatin of $(247 \pm 79)$ s for HT-RBPJ-WT to $(194 \pm 26)$ s (R/H), $(159 \pm 15)$ s (K/E), $(133 \pm 20)$ s (KRS/EHD), and $(142 \pm 12)$ s (RFL/HAA), respectively (Fig. 2f, Supplementary Figs. 9 and 10, and Supplementary Table 1-3). The mutations R/H, K/E and KRS/EHD additionally decreased the unspecific residence time of HT-RBPJ-WT (Fig. 2g). In contrast, the cofactor binding mutations FL/AA or absence of the

cofactor SHARP did not alter the average specific residence time of HT-RBPJ-WT (Fig. 2f), but increased the unspecific residence time (Fig. 2g). For several TFs, a correlation between residence time and repressive or activating function has been observed[1–8]. Similarly, our measurements of the residence times indicate that RBPJ variants with longer residence time show increased functional binding, as determined with the RBPJ-VP16 fusions (Fig. 2h). This suggests that the DNA residence time of a TF is an indicator of its functionality.

**The target site search time increases for DBD mutants of RBPJ**

We next quantified to which extent mutations in the DNA or cofactor binding interface altered the association of RBPJ to a specific target site at repressive conditions. In the mammalian nucleus, direct association of a TF to a specific target site is slow and association is rather thought to proceed via a faster search mechanism combining three-dimensional diffusion in the nucleoplasm and lower-dimensional exploration of the local environment[10,41]. Speed-up of the search process might occur via transient interactions including unspecific binding, sliding or hopping on DNA, association to cofactors, or nuclear structures such as protein clusters. We here denote any such mechanism as facilitated diffusion. Facilitated diffusion may entail a transition of the TF from transient, unspecific associations to specific binding, potentially associated with a conformational switch[42,43]. Structural data from DNA-bound vs unbound Su(H)[44], the RBPJ protein

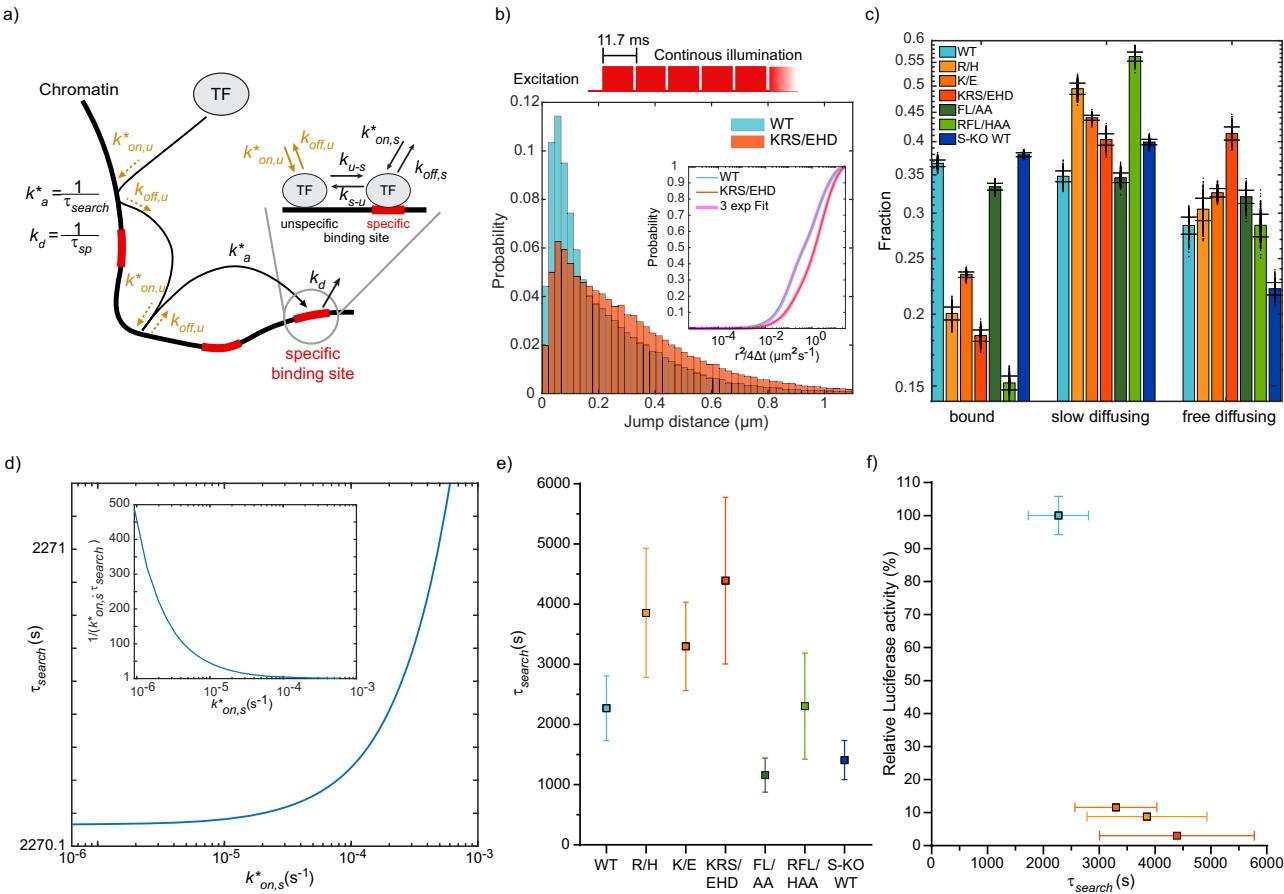

**Fig. 3 | Bound fractions and target site search times of HT-RBPJ variants.**
**a** Three-state model of the transcription factor target site search process via facilitated diffusion. Unbound transcription factors associate to a specific binding site with the overall association rate $k^*_a$ combining the two pathways of direct association with rate $k^*_{on,s}$ and of indirect association via unspecific binding with rate $k^*_{on,u}$ close to the specific site and sliding into specific binding with the transition rate $k_{u\text{-}s}$. Multiple unspecific binding events may occur before the specific site is bound. Dissociation from the specific binding site occurs with the overall dissociation rate $k_d$ combining the two pathways of direct dissociation with rate $k_{off,s}$ and indirect dissociation via transition to unspecific binding with rate $k_{s\text{-}u}$ and unspecific dissociation with rate $k_{off,u}$. **b** Upper panel: Scheme of illumination pattern for fast tracking with indicated frame cycle time. Lower panel: jump distance distribution of HT-RBPJ-WT (blue) and HT-RBPJ-KRS/EHD (orange). Inset: cumulative jump distance distributions with three-component diffusion model (pink).

**c** Fractions of the three-component diffusion model and assignment to bound, slow and fast diffusing molecules. Data represents mean values ± s.d. from 400 resamplings with randomly selected 80 % of the data. For experimental statistics see Supplementary Table 5. **d)** Target site search time ($\tau_{search}$) of HT-RBPJ-WT as function of the direct association rate ($k^*_{on,s}$). Inset: fold-acceleration of association via facilitated diffusion over direct association as a function of $k^*_{on,s}$. **e** Target site search time of HT-RBPJ variants at similar ratio of direct to overall dissociation, calculated from data in c), Fig. 2f, g, and Supplementary Table 1 and 4 (Methods). Data represented as value ± s.d. (Gaussian error propagation). Statistics are provided in Supplementary Tables 3 and 5. **f** Relative luciferase activity of HT-RBPJ-VP16 fusion variants versus their search time. Values denote mean ± s.d. (replot of data from Figs. 1b and 3e). Color code in **b**, **c**, **e**, **f** as in Fig. 1. Source Data are provided as a Source Data file for Fig. 3b, c, e, f.

in *D. melanogaster*, revealed distinct conformational changes and a reduction of disordered amino acids when bound to its target site (PDB: 5E24, Supplementary Fig. 11). While the search mechanism of RBPJ to find a specific target site is currently unknown, we therefore nevertheless assumed that it also follows a mechanism of facilitated diffusion.

Recently, we devised a three-state model of the search process by facilitated diffusion that relates the target site search time, given by the average time a TF molecule needs to find a specific target sequence among a myriad of transient unspecific interaction sites, to the transition rates between the unbound state and the unspecifically and specifically bound states[45] (Fig. 3a). Of note, the unspecifically bound state of the three-state model may represent any transient interaction, be it with DNA, cofactors or other nuclear structures. The target site search time $\tau_{search}$ is the inverse of the association rate $k_a$, which combines the pathways of direct association to the specific target site and indirect association via unspecific binding (Methods). It depends on the experimentally accessible parameters of the unspecifically ($p_u$),

specifically ($p_s$) and unbound ($p_f$) fractions of the TF, the dissociation rate $k_{off,u}$ from an unspecific site and the effective specific dissociation rate $k_d$. $k_d$ combines the pathways of direct dissociation from the specific target site and indirect dissociation via unspecific binding. In addition, one of the direct target site association ($k_{on,s}$) or dissociation ($k_{off,s}$) rates or of the microscopic transition rates $k_{u\text{-}s}$ or $k_{s\text{-}u}$ between unspecific and specific binding needs to be known to calculate the other missing parameters and the target site search time[45] (Methods).

To obtain the fraction of molecules bound both unspecifically and specifically, we measured the diffusive behavior of HT-RBPJ variants by recording continuous movies with 10 ms camera integration time (Fig. 3b, Supplementary Table 4 and 5, and Supplementary Movie 2). We collected the jump distances of detected single-molecule tracks in cumulative histograms and analyzed these with a diffusion model including slow, intermediate and fast diffusing molecules, which described data better than a two-component model (Fig. 3b, Supplementary Fig. 12 and Methods)[39,46]. The diffusion coefficients of HT-RBPJ variants were similar, compatible with their similar size (Supplementary

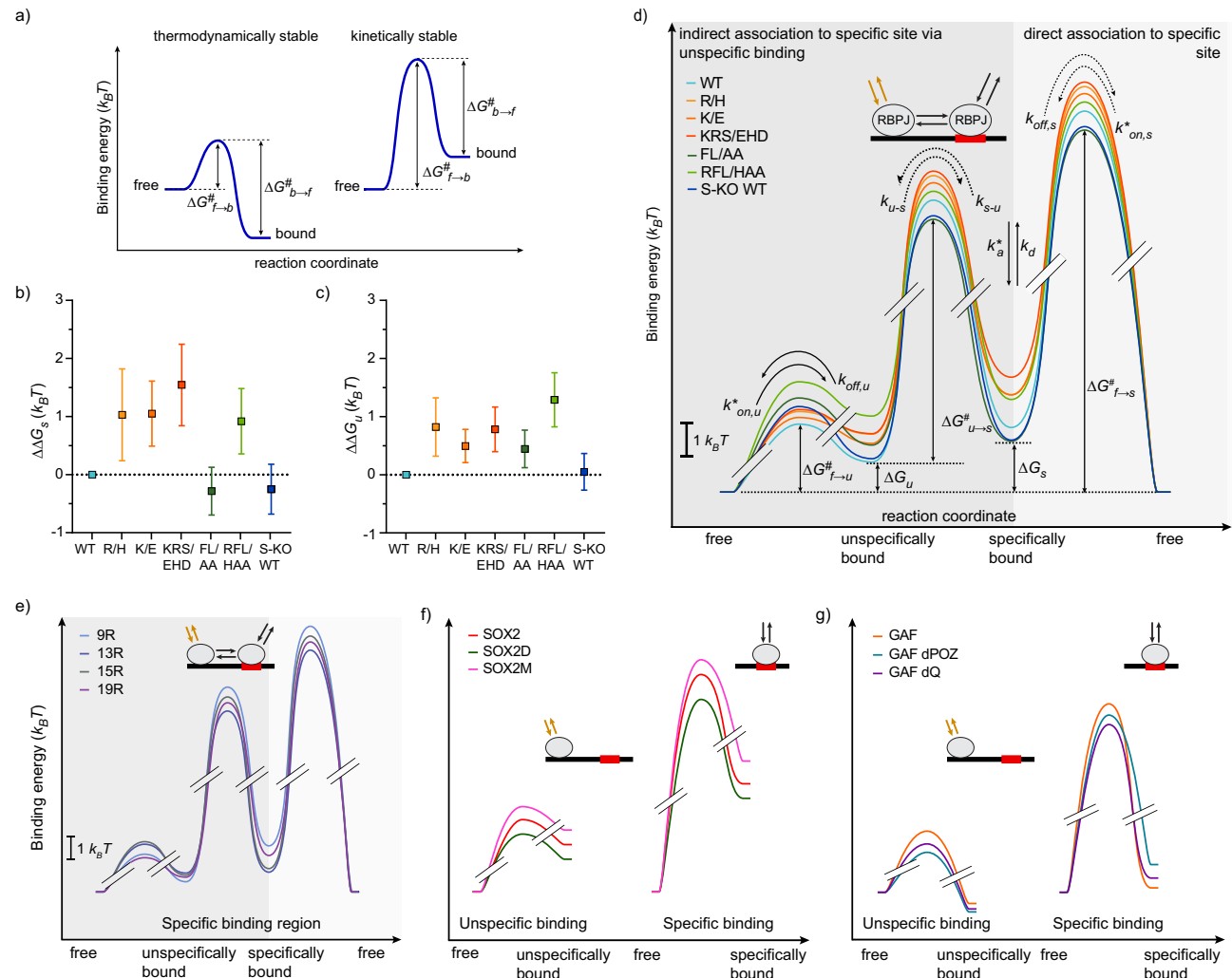

**Fig. 4 | In vivo binding energy landscape of transcription factors. a** Sketch of binding energy landscapes of a thermodynamically stable and a kinetically stable binding interaction. In both cases, the energy barrier of unbinding and thus the residence time in the bound state is equal. Binding energy differences. Differences in **b** specific binding energies $\Delta\Delta G_s$ and **c** unspecific binding energies $\Delta\Delta G_s$ between HT-RBPJ-WT and mutants, calculated from data in Fig. 2f, g, Fig. 3e and data listed in Supplementary Table 7. Data represented as value ± s.d. (Gaussian error propagation). Statistics are provided in Supplementary Tables 3 and 5. **d** In vivo chromatin binding energy landscapes of HT-RBPJ variants. Energy differences between bound states or transition barriers and the free state as well as kinetic rates of the target site search process by facilitated diffusion are indicated. Dark gray shade: specific association via unspecific binding and subsequent transition to specific binding; bright gray shade: direct specific association. Only relative, but not absolute energy differences of transition barriers can be obtained. Dotted arrows indicate undetermined rates, limits of the relative transition barriers are given in Supplementary Fig. 9. In vivo chromatin binding energy landscapes of **e** TALE variants[2], **f** SOX2 variants constructed from published kinetic rates[14], and **g** GAF variants constructed from published kinetic rates[15]. Color code in **b**–**d** as in Fig. 1. Source Data are provided as a Source Data file for Fig. 4b–g.

Fig. 13a). The slow diffusion component arises due to slow movement of chromatin bound by HT-RBPJ molecules and the single-molecule localization error. It represents the fraction of molecules bound to either unspecific or specific chromatin sites[47–49]. We found a bound fraction of ~37% for HT-RBPJ-WT, similar to the value of 34% observed in *D. melanogaster* salivary gland cells[50]. Overexpression of RBPJ variants compared to endogenous RBPJ might lead to saturation of binding sites, thereby artificially decreasing the observed bound fraction. However, a linear relation between bound molecules and all detected molecules confirmed that binding was not saturated in our overexpression conditions (Supplementary Fig. 8e). Thus, the bound fraction should largely be unaffected at our levels of overexpression. As expected, the overall bound fraction was reduced to ~20% (R/H), ~23% (K/E), and ~18% (KRS/EHD) if the DNA binding interface of RBPJ was disturbed (Fig. 3c and Supplementary Table 4 and 5). Moreover, the fraction of specifically bound HT-RBPJ molecules correlated with the number of target sites

identified by ChIP-Seq, similar to previous findings (Supplementary Fig. 13b)[51].

We calculated the target site search time $\tau_{search}$ of HT-RBPJ-WT by combining the results of our residence time and binding fraction measurements and varying the kinetic rates $k^*_{on,s}$, $k_{off,s}$, $k_{u-s}$ and $k_{s-u}$ that were not experimentally accessible (Methods). This yielded a minimal target site search time of ~2270 s, which increased in a small interval up to ~2271 s for a variation of $k^*_{on,s}$ over three orders of magnitude (Fig. 3d, Supplementary Fig. 14, and Supplementary Table 6). At the upper limit of $\tau_{search}$, the time of direct association equaled the target site search time, and a description of the search mechanism by facilitated diffusion starts to break down.

To compare the target site search times of different HT-RBPJ variants, we kept the ratio of the direct dissociation rate $k_{off,s}$ to the measured effective dissociation rate $k_d$ constant for all HT-RBPJ variants. All mutations in the DNA binding interface increased the target site search time of HT-RBPJ-WT (Fig. 3e and Supplementary Table 7). In contrast, the

mutations FL/AA disturbing cofactor binding or the absence of SHARP reduced the search time. Interestingly, our measurements of the search times indicate that RBPJ variants with shorter search time show increased functional binding, as determined with the RBPJ-VP16 fusions (Fig. 3f). Thus, in addition to the DNA residence time, the target site search time is an indicator of RBPJ function. Together, both parameters determine the occupancy of the target sites.

**RBPJ function is governed by association and dissociation rates**
In a binding reaction at equilibrium, the effective association and dissociation rates of the reactants are coupled through the equilibrium constant and define the binding energy of the reaction (Fig. 4a and Methods). If the effective association rate is larger than a given dissociation rate, the energy of the bound state is lower than the energy of the unbound state, and the reaction is thermodynamically stable (Fig. 4a). If the effective association rate is smaller than the given dissociation rate, the energy of the bound state is higher than the energy of the unbound state. Importantly, in this case, the interactions underlying the adhesion of both reactants are still present, and the reactants are kinetically trapped in the bound state, making it kinetically stable.

To illustrate the effect of mutations in the DNA and cofactor binding interfaces of RBPJ on the RBPJ binding kinetics in the nucleus, we propose to construct the effective in vivo binding energy landscape of RBPJ. It can be obtained from its unspecific and specific bound fractions, the effective association rate, and the dissociation rate (Methods). Importantly, while TF-DNA binding free energies are typically compared at standard concentrations of binding sites, taking the cellular concentrations of binding sites of the TFs into account enables comparing the situation the TFs actually encounter in the nucleus.

We first determined the effective in vivo binding energy $\Delta G_s$ of specific binding, that is the effective energy difference between binding to a specific target sequence and the unbound state, from the ratio of the specifically bound to the unbound fraction of molecules for each of the HT-RBPJ variants (Supplementary Table 8 and "Methods"). The contribution of a mutation to the specific effective binding energy of RBPJ is then contained in the energy difference $\Delta\Delta G_s$ with HT-RBPJ-WT (Fig. 4b). Consistent with compromised DNA binding, all mutations in the DNA binding interface, R/H, K/E, KRS/EHD, and RFL/HAA increased the specific effective binding energy of HT-RBPJ-WT. In contrast, mutation FL/AA disturbing cofactor binding or absence of SHARP slightly decreased $\Delta G_s$ compared to HT-RBPJ-WT. Thus, the specific effective binding energies of HT-RBPJ variants illustrate mutation-specific changes in the RBPJ chromatin binding stability in the nucleus.

Similar to $\Delta G_s$ of specific binding, we determined the unspecific effective binding energy $\Delta G_u$ from the ratio of the unspecifically bound fraction to the unbound fraction of HT-RBPJ variants (Fig. 4c) (Methods). Again, all mutations in the DNA binding interface increased the unspecific effective binding energy of HT-RBPJ-WT.

We further calculated other basic points of the effective in vivo binding energy landscape of HT-RBPJ-WT and mutants: in addition to $\Delta G_s$ and $\Delta G_u$ also the effective energy barriers $\Delta G^{\#}_{f\to u}$ and $\Delta G^{\#}_{f\to s}$ of the direct unspecific and specific association and dissociation processes, and the effective energy barrier $\Delta G^{\#}_{u\to s}$ of the transition from unspecific to specific binding at a target sequence (Fig. 4d and Supplementary Table 8). We again used the ratios of bound to unbound fractions as well as the kinetic rates obtained from the three-state model of the target site search process to calculate these effective energy barriers (Methods). While we could not obtain the absolute height of the barriers without knowledge of the frequency factor, the relative height of barriers can be compared. The binding energy landscape of HT-RBPJ-WT was comparable in the presence and absence of endogenous RBPJ (Supplementary Fig. 15).

A striking feature of the effective in vivo binding energy landscapes of HT-RBPJ variants is that the specifically bound state has a higher energy than the unspecifically bound and the unbound, free state, $\Delta G_s > \Delta G_u > 0$ (Fig. 4d). Counterintuitively, this means that binding of RBPJ to a specific target sequence is thermodynamically unstable in vivo, whereas unspecific binding and even more the free state are thermodynamically favored. This arises from the longer association time than residence time of unspecific and specific binding, and is reflected in the small bound fractions below 0.4 of HT-RBPJ variants (Figs. 4a and 3c). Yet we emphasize that binding of RBPJ to chromatin is kinetically stable, with residence times of up to ~250 s for HT-RBPJ-WT.

**Transcription factor binding is kinetically not thermodynamically stable in vivo**
Encouraged by the insight obtained from the effective in vivo binding energy landscape of RBPJ, we reanalyzed our previously measured live-cell single molecule data of TALE (transcription activator like effector) TFs (TALE-TFs)[2]. The TALE-TFs differed in the length of their DNA binding domain and showed varying residence times on chromatin[2]. We now additionally calculated the basic points of the effective binding energy landscape of TALE-TFs from the bound fractions and kinetic rates analogous to RBPJ (Fig. 4e and Supplementary Table 9). Similar to RBPJ, binding of TALE-TFs to chromatin is not thermodynamically but kinetically stable in vivo. Moreover, we calculated the effective binding energy landscapes of SOX2 and GAF and of various DNA binding domain- or activation domain-mutants of these factors using published kinetic rates[14,15] (Fig. 4f, g, Supplementary Table 10). Analogous to RBPJ and TALE-TFs, specific chromatin binding of SOX2 and GAF occurred at higher effective energy than unspecific binding and the unbound state and thus was thermodynamically unstable in vivo. This indicates that kinetic instead of thermodynamic stability of chromatin binding might be a general feature of transcription factors in vivo.

## Discussion
AOS is linked to the autosomal dominant mutation K195E (K/E) in the DNA binding interface of RBPJ[27,28]. Our live-cell measurements showed that HT-RBPJ-K/E still binds to DNA and is able to activate gene transcription in presence of NICD, albeit to a lower degree than HT-RBPJ-WT (Fig. 1b, c). In particular, we found that mutation K/E increased the target site search time of RBPJ 1.5-fold, reduced the residence time of RBPJ at specific target sites by ~36% and increased the effective binding energy by ~1 $k_BT$, corresponding to a 2.1-fold smaller specifically bound fraction. ChIP-Seq experiments confirmed reduced chromatin binding of HT-RBPJ-K/E, and revealed a loss of 47% of the HT-RBPJ-WT target sites (Fig. 1e). Our findings allow refining possible molecular mechanisms underlying AOS: If one allele carries the mutation, the amount of wild-type RBPJ will be reduced, thus reducing the occupation frequency of target sites exclusively bound by RBPJ-WT. This reduced target site occupation and the shorter residence time and lower transcription activation potency of RBPJ-K/E at sites bound by both species should limit the repression and activation of genes, in addition to proposed cofactor sequestration[28]. Moreover, off-target binding of HT-RBPJ-K/E as revealed by ChIP-Seq (Fig. 1g) might lead to a gain-of-function phenotype. This scenario could be similar as observed for point mutations of IRF4 associated with autosomal dominant combined immunodeficiency[52,53]. Overall, our data predict de-repression and lower than normal Notch-activated transcription of numerous Notch target genes in AOS.

For HT-RBPJ-WT and mutant variants, we observed binding events stretching over several orders of magnitude up to hundreds of seconds (Fig. 2c). This is in accordance with survival time distributions reported for other TFs including the glucocorticoid (GR) and estrogen (ER) receptors[8,54], SRF[7], CDX2[40], FoxA1[55], CTCF[55], TBP[56], and GAF[15]. Since even the bacterial TFs LacI and TetR, which lack specific target sequences in mammalian cells, exhibited such long survival times[57,58], and functional low-affinity binding sites had been described[59], it was

argued that specific and unspecific binding cannot be classified by residence time[60,61]. However, by combining the readout for functional binding (Fig. 1b), binding characterization by ChIP-Seq (Fig. 1d, e), and differences in the dissociation rate spectra of wild type and mutant RBPJ (Fig. 2d), we were able to identify the time regime of specific binding of RBPJ variants. On average, specific binding lasted longer than hundred seconds. Nevertheless, we cannot exclude that short binding events of RBPJ also have functional consequences.

To obtain the search times of RBPJ variants, we were limited by the unknown search mechanism of RBPJ. We therefore employed a three-state model including a diffusive state, a state of transient unspecific interactions and a state of specific binding to DNA[45] (Fig. 3a), which is general enough to capture the coarse features thought to speed up the target search of TFs in mammalian cells[10,41]. Thus, the model should be a good starting point to quantify the search time of RBPJ. Future studies will be necessary to provide more comprehensive mechanistic insight into the search process of RBPJ.

From our measured dissociation rates and number of binding sites and the calculated association rate, we can estimate the affinity of HT-RBPJ-WT. If we assume a nucleus with an elliptical volume of $4/3*\pi*12*8*3\ \mu m^3$, we obtain the dissociation constant $K_d = k_d\ [binding\ sites]/k^*_a = 191$ nM. This in vivo affinity is comparable to the affinity of 74 nM measured in vitro by isothermal titration calorimetry for a standard RBPJ binding site[28]. Since our binding measurements average over all binding sites in the nucleus, a smaller affinity is expected. The comparable in vivo and in vitro affinities give further support to our approach to obtain kinetic rates by live-cell single molecule tracking.

Reconstructing the effective binding energy landscapes of various RBPJ variants and other TFs from their bound fractions and association and dissociation rates revealed that TF-DNA binding is thermodynamically unstable in vivo, while the slow association and dissociation rates of the TFs ensure kinetic stability. While the residence time is mostly fixed within the structure of the TF, the target site search time can be adjusted by varying the concentration of TFs. Thus, by expressing more TFs in response to internal or external stimuli, the cell is able to decrease the TF search time and thereby shift the binding equilibrium towards more favorable binding. In case the search time becomes shorter than the residence time, binding becomes thermodynamically stable and bound fractions can increase above 0.5. The TF concentrations found in the cell presumably reflect the need to balance sufficiently high TF binding with minimal TF concentrations to economize limited resources.

The effective in vivo binding energy landscape of RBPJ reveals important insight about the mechanism of action of RBPJ mutations (Fig. 4d). The DNA binding mutations R/H, K/E, and KRS/EHD weakened both specific and unspecific binding. Binding was destabilized not only by increasing the direct dissociation rates $k_{off,s}$ and $k_{off,u}$, but also by decreasing the direct association rates $k^*_{on,s}$ and $k^*_{on,u}$. This resulted in an increase of both the transition barriers $\Delta G^{\#}_{f\rightarrow s}$ and $\Delta G^{\#}_{f\rightarrow u}$ and the energies $\Delta G_s$ and $\Delta G_u$ of specific and unspecific binding. Interestingly, changes in the barrier $\Delta G^{\#}_{f\rightarrow s}$ of specific binding were mirrored in the changes of the barrier $\Delta G^{\#}_{u\rightarrow s}$ to switch from unspecific to specific binding.

In contrast to mutations in the DNA binding interface, the mutations FL/AA in the cofactor binding interface or absence of SHARP mainly increased the barrier $\Delta G^{\#}_{f\rightarrow u}$ to unspecific binding, while the barriers to specific binding were decreased by a faster association rate, resulting in a lower effective binding energy $\Delta G_s$ (Fig. 4d). The mutations FL/AA impede interaction of RBPJ with cofactors including SHARP, RITA1 and NICD[22,25,62]. As expected, the NICD-mediated transcription activation potency of HT-RBPJ-FL/AA was reduced (Fig. 1c). We further observed that HT-RBPJ-FL/AA loses the ability to associate to many of the HT-RBPJ-WT target sites, despite an intact DNA binding domain (Fig. 1e). Similarly, HT-RBPJ-WT showed fewer binding events in the absence of SHARP (Fig. 1e). Our findings point towards the

possibility that cofactors of RBPJ might contribute in guiding RBPJ to its cognate target sites, similar to the role of the activation domain of several TFs in determining DNA specificity[63,64].

Our approach to determine the effective in vivo binding energy landscape by measuring bound fractions and kinetic rates using single-molecule tracking is model-free. It is therefore not limited to protein–DNA interactions. The model that we additionally employed here for the search process of TFs enabled revealing more details of TF–DNA interactions such as the transition from unspecific to specific binding at the target sequence, but is generally not required. Similar to in vivo measurements of protein–protein dissociation constants by single-molecule tracking[65], fluorescence cross-correlation spectroscopy[66–68] or Förster resonance energy transfer[69], our single-molecule approach including kinetic rates should be readily applicable to other biomolecular interactions in vivo.

To conclude, our measurements uncovered insight into the association and dissociation kinetics and the cofactor-dependance of RBPJ-DNA binding, and the molecular mechanism of the disease-related mutation K195E. It will be an important task for the future to develop models of gene regulation including slow target site search kinetics of TFs in non-equilibrium conditions[70]. Moreover, kinetic rather than thermodynamic stability might also be valid for protein-protein and other biomolecular interactions in vivo, if association is slow compared to dissociation.

## Methods

### Cell culture
HEK293 (ATCC: CRL-1573) and HeLa (ATCC: CCL2) derived cell lines were grown in DMEM additional supplemented with 10% fetal bovine serum, 1% glutamax, 1% non-essential amino acids and 1% sodium pyruvate and cultured at 37 °C and 5% $CO_2$.

### Transient DNA transfection
To transiently transfect HeLa^RBPJ KO cells (clone #42) with expression plasmids and reporter plasmids (expression plasmids are listed in Supplementary Table 11) for subsequent luciferase assays, we used the Lipofectamine 2000 transfection reagent (Invitrogen, Cat.No.: 11668019) according to the manufacturer's instructions.

### Luciferase assay
To determine the luciferase activity of previously transfected HeLa^RBPJKO cells, we used the luciferase assay system from Promega (Promega, Cat. No.: E1501). To do so, we seeded $2.25 \times 10^4$ cells per well onto a 48-well plate and transfected them after 24 h with 0.25 μg reporter plasmid per well. Furthermore, were co-transfected some cells with 100 ng of expression plasmids for RBPJ mutants. For cell lysis, we applied 100 μl of 1:5 diluted cell culture lysis reagent (Promega, Cat. No.: E1531) to each well, 24 h after transfection. We centrifuged the resulting cell lysates and applied 10 μl of the supernatant to a 96-well plate. Finally, we used a Luminometer Microplate reader LB960 (Berthold Technologies, Cat. No: S11902) in order to determine the luciferase activity. We performed at least four independent experiments with two technical replicates.

### Co-immunoprecipitation
To verify protein-protein interaction between Halo-tagged RBPJ proteins and Flag-tagged NICD and Flag tagged SHARP (aa 2770-3127)[20,28] we performed Co-Immunoprecipitation experiments. For Co-immunoprecipitation of Halo-tagged RBPJ proteins together with Flag-tagged NICD we co-transfected HEK293 cells with expression plasmids for the given proteins (used plasmids are listed in Supplementary Table 12). Cells were lysed 24 h after transfection using 600 μl CHAPS lysis buffer. We incubated 500 μl CHAPS lysate with agarose beads conjugated with anti-Flag antibodies (M2, Merck, Cat. No.: A2220) overnight at 4 °C on an overhead shaker. Samples were washed

six times with CHAPS lysis buffer and resuspended in 1x Laemmli buffer for subsequent SDS-PAGE.

## Generation of CRISPR/Cas9 depleted cells

We designed the CRISPR/Cas9 guides using the online tool available at http://crispor.tefor.net/. We added the desired 5' overhangs and phosphorylated, annealed and ligated the oligos into the px459 v2.0 (Addgene #62988)[71] predigested with BbsI. The *SHARP* depletion was generated with the combination of hSHARP guide #1 and hSHARP guide #2. The *RBPJ* depletion was generated with the combination of hRBPJ guide #1 and hRBPJ guide #2 (sequences of the guides in Supplementary Table 13). We transfected HeLa cells (ATCC CCL-2) with 10 μg of each px459 v2.0 plasmid together with 40 μl of linear PEI (Polyscience 23966) using standard protocols. We selected the cells with puromycin before establishing single cell clones.

## RNA extraction, reverse transcription, quantitative PCR (qPCR)

To purify the total RNA, we used the RNeasy Mini Kit (Qiagen #74104), the QIAshredder (Qiagen #79654) and the DNase I (Qiagen #79254) accordingly to the manufacturer´s instructions. For generation of cDNA, we used 1 μg of RNA and retro-transcribed it using random primers and SuperScript II reverse transcriptase (Invitrogen #18064-014). We assembled qPCR reactions with QuantiFast SYBR Green RT-PCR Kit (Qiagen #204156), gene-specific oligonucleotides (Supplementary Table 14) and analyzed using the LightCycler480™ system (Roche Diagnostics). We calculated mRNA expression levels relative to the housekeeping gene *Hypoxanthine Phosphoribosyltransferase 1* (*HPRT*).

## ChIP-Seq

Cells (HeLa) were washed twice with PBS, fixed for 1 h at room temperature in 10 mM dimethyladipimate (DMA, Thermo Scientific 20660), dissolved in PBS and after washed once in PBS crosslinked in 1% FMA for 30 min at room temperature. We blocked the FMA reaction by adding 1/8 volume of 1 M glycine pH 7.5 and incubated for 5 min at room temperature. We lysed cells in 1 ml of SDS Lysis Buffer (1% SDS, 10 mM EDTA, 50 mM Tris-HCl pH 8.1) and sonicated using the Covaris System S220 AFA (28 cycles, 30 s ON, 30 s OFF). We diluted the chromatin with ChIP Dilution Buffer (0.01% SDS, 1.1% Triton X-100, 1.2 mM EDTA, 16.7 mM Tris-HCl pH 8.1, 167 mM NaCl) and pre-cleared with protein-A-Sepharose beads (GE Healthcare 17-5280-02) for 30 min at 4 °C. The chromatin was subsequently incubated over night with the proper amount of the desired antibody. We immunoprecipitated the chromatin with anti-RBPJ antibody (Cell Signaling Technology, 5313S) and after washing, we eluted the chromatin with Elution Buffer (50 mM Tris pH 8.0, 10 mM EDTA pH 8.0, 1% SDS). After reverting the crosslinks, we purified the DNA with phenol/chloroform/isoamyl alcohol and the Qiaquick PCR purification kit (Qiagen, 28104) and eluted in H$_2$O[72]. For spike-in purposes, we used chromatin from *D. melanogaster* Schneider cells in presence of 2 μg of anti-His2Av antibody (Active Motif 61686) for each immunoprecipitation. Antibodies are listed in Supplementary Table 15.

We prepared libraries using the Diagenode MicroPlex Library Preparation kit v3 (Diagenode C05010001) following the manufacturer's instructions with few modifications. Subsequently, we purified libraries with Agencourt AMPure XP Beads (Beckman Coulter, #A63881), and quantified and analyzed on an Agilent Tapestation device. Finally, we performed sequencing on a NovaSeq device at Novogene UK.

## ChIP-Seq analysis

We applied quality and adapter trimming to raw sequencing reads with TrimGalore (https://github.com/FelixKrueger/TrimGalore). Next, we aligned the trimmed reads to the human reference genome (hg19, downloaded from Illumina's iGenomes) using Hisat2[73] with "-k 1 –no-spliced-alignment –phred33" parameters and stored them as binary alignment maps (BAM). To filter BAM files for PCR duplicates, we used the *MarkDuplicates* function of the Picard tools (available at http://broadinstitute.github.io/picard/) with "REMOVE_SEQUENCING_DU-PLICATS = true REMOVE_DUPLICATES = true" parameters. Next, we generated normalized coverage tracks (bigWigs) using the Deeptool's *bamCoverage* function[74] based on the filtered BAM files. To call peaks for the individual samples, we used PeakRanger[75] with p- and q-values cutoffs of 0.0001 and the matching input for each sample. We selected only peaks that were detected in both replicates and not overlapping with ENCODE's blacklisted regions (https://github.com/Boyle-Lab/Blacklist/) for further analysis. By using IGV[76], we visually inspected the conclusiveness of the peak calling. Subsequently, we performed motif analysis using the MEME-suite[77] based on the summits of RBPJ peaks detected by PeakRanger ±50 base pairs. We generated heat maps using Deeptools's *computeMatrix* and *plotHeatmap* functions based on the RBPJ peak set detected in HeLa control cells and the normalized coverage tracks for all samples. To generate Snapshots for example regions, we used Gviz[78].

## Stable cell line generation

We thawed LentiX-293T packaging cells at least 1 week prior to the virus production. To start the virus production, we transfected LentiX cells with 3 different plasmids, the plasmid pLV-tetO-Oct4 (Addgene #19766)[79] including the sequence of interest, the packaging plasmid psPAX2 (Addgene #12260) and the virus envelope plasmid pMD2.G (Addgene #12259) using JetPRIME (Polyplus-transfection, #114-15). To obtain a sufficient amount of virus, we let LentiX cells grow to 80 % confluency on a 10 cm culture dish before transfection. Next, we mixed 10 μg of plasmid encoding for the Halo-tagged construct of interest, 7.5 μg psPAX2, and 2.5 μg of pMD2.G with 500 μl of jetPRIME buffer and briefly vortexed the resulting mix. We added 30 μl of jetPRIME transfection reagent to the mixture and shortly vortexed. Subsequently, we incubated the transfection mixture for 10 min at room temperature. After incubation, we added the mixture dropwise to the LentiX cells and incubated them for 2 days at 37 °C and 5% CO$_2$.

One day before lentiviral transfection, we seeded 40,000 Hela cells in the wells of a 6-Well plate. We harvested the viruses by collecting the supernatant (9–10 ml) of LentiX cells with a syringe, filtering it through a Whatman™ 0.45 μm membrane filter and collecting the flow-through in a 15 ml falcon tube. After that, we added 1 ml of filtered virus solution to the Hela cells followed by an incubation for 3 days at 37 °C and 5% CO$_2$. After 3 days, we expanded the cells on a 10 cm culture dish by washing them once with 1 ml PBS and trypsinizing them with 0.5 ml 0.05% Trypsin for 4 min at 37 °C and 5% CO$_2$. We stopped trypsinization by adding 4.5 ml DMEM.

To test whether the viral transduction was successful, we labeled the cells with 2.5 μM HaloTag-Ligand (HTL) TMR (Promega #G8251) for 15 min, following the manufacturer's protocol, and observed them under a confocal spinning disk microscope. In order to obtain a highly homogenous positive cell population, we sorted the cells via their fluorescence using FACS. For this, we labeled cells with 1.25 μM HTL-TMR following the manufacturer's protocol. To adjust the flow cytometer gates, we used unlabeled Hela cells.

## Western blotting

To verify the stable expression of target proteins in the aforementioned stable cell lines, we performed western blots in order. Therefore, we lysed cells by applying 120 μl of CHAPS lysis buffer containing protease inhibitors to cell pellets. After an incubation on ice for 1 h, we centrifuged the samples for 30 min at 21,500 × *g* (Eppendorf Refrigerated Centrifuge 5417R) and 4 °C. Subsequently, we collected the supernatant in a new 1.5 ml reaction tube and determined the protein concentration of each sample by a Bradford protein assay (Biorad, Cat.

No.: 5000006). For each sample, we mixed 20 μg of protein with 6x Laemmli buffer and applied it to a 10% SDS polyacrylamide gel. After gel electrophoresis, we blotted the proteins at RT on a PVDF membrane (Merck, Cat. No.: IPVH00010). We blocked the membranes for 1 h at RT with sterile filtered 5% bovine serum albumin (BSA) (Serva, Cat. No.:9048-46-8) dissolved in TBS with 0.1% Tween-20, prior to incubation with the primary anti-Halo-Tag antibody (Mouse monoclonal antibody, Promega, Cat. No.: G9211). For incubation with the primary anti-RBPJ antibody (Rat monoclonal antibody, Cosmo Bio, Cat. No.: SIM-2ZRBP2), we blocked the membranes under the same condition as above using 5% skim milk (PanReac AppliChem, Cat. No.: A0830) dissolved in PBS with 0.1% Tween-20. To stain with the anti-Halo-Tag antibody, diluted 1:500 in TBS-T, we incubated the membranes over night at 4 °C. To apply the anti-RBPJ antibody, we used PBS-T for a final dilution of 1:1000. After washing the membranes three times with PBS-T to remove unbound primary antibodies, we used horseradish peroxidase conjugated secondary antibodies against mouse (GE Healthcare, Cat. No.: NA931V) or rat (Jackson ImmunoResearch, Cat. No.: 112-035-071) to detect primary antibodies. To detect the target proteins, we incubated the membranes for 1 min in 5 ml of ECL solution (Cytiva,Cat. No.: RPN2209). We detected the resulting chemiluminescence signal with High performance chemiluminescence films (Cytiva, Cat. No.: 28906837). All antibodies used are shown in Supplementary Table 16.

## Quantification of Western blot data

We performed Western blots using five individual lysates of wild-type HeLa cells stably expressing HT-RBPJ-WT to quantify the expression of HT-RBPJ-WT proteins relative to endogenous RBPJ. We determined expression levels by quantifying the mean intensities of the bands of endogenous RBPJ and HT-RBPJ-WT using ImageJ (https://imagej.net/ij/). The expression levels of HT-RBPJ-WT are calculated relative to endogenous RBPJ. Finally, we calculated the mean of all individual relative values to determine the final relative expression level of HT-RBPJ-WT. The relative co-immunoprecipitation (co-IP) efficiencies (Supplementary Fig. 2) were calculated by dividing the band intensities of HT-RBPJ variants by the band intensities of Flag-NICD or Flag-SHARP. Finally, we calculated the average of all individual relative values and normalized to the average value of HT-RBPJ-WT to determine the final relative co-IP efficiencies.

## Flow cytometry to determine the cellular abundance of HT-RBPJ-WT

To determine the average amount of HT-RBPJ-WT proteins in the HeLa cell line stably expressing HT-RBPJ-WT, we performed flow cytometry (FCM) measurements with an Attune NxT flow cytometer and compared the intensity level with a calibrated reference U2OS cell line stably expressing Halo-tagged CTCF (C32)[80]. We calibrated the device and determined the background fluorescence using untreated HeLa and U2OS cells. We seeded the cells in 10 cm dishes and stained them on the day of the FCM measurement at a confluence of 80 %. To stain the cells, we incubated them for 30 min at 37 °C with 500 nM Halo-TMR ligand. Subsequently, we aspirated the medium and washed the cells once with PBS. After removing PBS, we added fresh pre-warmed medium before detaching the cells by trypsinization. We transferred the cell suspension into fresh DMEM and determined the cell concentration using a Neubauer hemocytometer. Next, we centrifuged the samples for 5 min at $1300 \times g$ (Centrifuge Beckman Coulter Allegra X-15-R, Rotor Sx4750A) at 4 °C. Prior to subjecting the cells to FCM, we aspirated the supernatant and resuspended the cells stored on ice in 220 μl PBS.

To determine the average amount of expressed HT-RBPJ-WT in HeLa cells, we calculated the relative fluorescence intensity by subtracting the background fluorescence and compared it to the reference fluorescence intensity of U2OS C32 cells (Supplementary Fig. 15). We estimated a mean protein abundance of 90,863 HT-RBPJ-WT molecules per cell. Since HT-RBPJ-WT was over-expressed 0.88-fold compared to endogenous RBPJ (Supplementary Fig. 8c), this corresponded to an average of 102,855 endogenous RBPJ molecules and a total number of 193,718 HT-tagged or endogenous RBPJ molecules per cell. Assuming an ellipsoidal nucleus with volume $\pi/6{*}8{*}8{*}5\ \mu m^3$, the corresponding average RBPJ concentration was ~1.92 μM.

## Widefield fluorescence microscopy

We used fluorescence microscopy to investigate the subcellular localization of the stably expressed HT-RBPJ variants. Therefore, we seeded $4 \times 10^4$ cells of each investigated stable cell line into one well of a 2-well chamber glass coverslip (Nunc LabTek, Cat. No.: 155380) that was previously incubated with a 1x fibronectin solution (Sigma-Aldrich, Cat. No.: F2006) for 30 min at 37 °C. Afterwards, we aspirated the fibronectin solution and subsequently washed the chamber coverslips twice with PBS. After 24 h, we stained the cells with 1 ml of a 1:2000 TMR Halo-ligand solution (Promega, Cat. No.: G8251) to fluorescently label the Halo-tagged proteins. After an incubation time of 15 min at 37 °C, we removed the Halo-ligand solution and washed the chamber coverslips three times with fresh DMEM medium, followed by an additional application of 1 ml of DMEM and a subsequent incubation of 30 min at 37 °C. We fixed the cells and labeled the DNA by incubating the cells with 1 ml of a DAPI solution (1:10,000 in PBS) on a shaker at RT. Finally, we washed the samples 5 times for 5 min with PBS and added one drop of fluoromount-G mounting medium (SouthernBiotech, Cat. No.: 0100-01) to each well prior to applying a cover slip.

Alternatively, in order to confirm the SHARP knock-out in HeLa SHARP knock-out clones #36 and #30, we used immunofluorescence. We applied 1 ml of DMEM, containing 45,000 cells, on coverslips and incubated them overnight at 37 °C. On the next day, we aspirated the medium and washed the coverslips once with PBS. After fixing the cells, we permeabilized them by applying 1 ml of 0.2 % Triton for 2 min. After two washing steps with PBS, we blocked unspecific binding of antibodies using PBS that contained 1% BSA, 1 % FBS, and 0.1 % fish-skin gelatine for 30 min at RT. We diluted the primary antibody (anti-SHARP.1[20]) 1:500 in blocking buffer and incubated the cells for 3 h. After five washing steps with PBS, we applied the secondary antibody for 1 h as a 1:1000 dilution in blocking buffer. After 30 min, we applied DAPI for a final dilution of 1:20,000, which was later removed with the secondary antibody. After 6 final washing steps with PBS, we added a drop of fluoromount-G mounting medium (SouthernBiotech, Cat. No.: 0100-01) on an object slide and placed the coverslip on top of it. Specifications of the used antibodies are shown in Supplementary Table 17. To image the cells, we used an Olympus IX71 fluorescence microscope, a 100 W mercury lamp (Osram, HBO 103 W/2) and a digital camera (Hamamatsu, C4742-95). To detect TMR (Excitation: ET545/25, Emission: ET 605/70), we used a Cy3 ET filter set (AHF, Cat. No.: F46-004), a suitable filter set for DAPI detection (Excitation: D360/50, Emission: D460/50 and an EGFP ET filter set (Excitation: ET470/40, Emission: ET 525/50) (AHF, Cat. No.: F46-002) for detection of green fluorescence.

## Preparation of cells for single-molecule imaging

One day prior to single-molecule imaging, we seeded cells on a 35 mm heatable glass bottom dish (Delta T, Bioptechs). For time-lapse imaging, we stained the cells with 3–6 pM HTL-SiR on the day of imaging. For that, we incubated cells with HTL-SiR for 15 min at 37 °C and 5% $CO_2$ and washed them once with PBS followed by a recovery step of 45 min at 37 °C and 5% $CO_2$ in DMEM. Directly before imaging, we washed cells three times with PBS and added OptiMEM for imaging.

To record continuous 11.7 ms single-molecule movies, we stained the cells for 1 h with 10 nM HTL PA-JF-646. After incubation, we washed the cells two times with PBS and incubated them further in DMEM for 45 min at 37 °C and 5% $CO_2$. Before imaging, we washed the cells three times with PBS and imaged them in OptiMEM.

## Single-molecule microscope setup

We conducted single-molecule imaging on a custom-built fluorescence microscope[81]. It is built around a conventional Nikon body (TiE, Nikon) equipped with an AOTF (AOTFnC-400.650-TN, AA Optoelectronics), a high-NA objective (100×, NA 1.45, Nikon), a 638 nm laser (IBEAM-SMART-640-S, 150 mW, Toptica) and a 405 nm laser (Laser MLD, 299 mW, Solna, Sweden). For a good signal-to-noise ratio, we illuminated cells with a highly inclined and laminated optical sheet (HILO)[38]. To detect the emitted fluorescence light, which previously passed a multiband emission filter (F72-866, AHF, Tübingen, Germany), we used an EMCCD camera (iXON Ultra DU 897, Andor, Belfast, UK).

## Single-molecule imaging

To assess the chromatin residence time of Halo-tagged molecules and to differentiate between unbinding of molecules and photobleaching, we performed time-lapse (tl) microscopy. Therefore, we recorded movies using a tl-cycle, which consisted of an image with a fixed camera integration time of 51.7 ms followed by a certain dark time. The tl-cycle times were 0.1 s, 0.4 s, 1.6 s, and 6.4 s. Overall, movies covered 30 s, 120 s, 480 s, and 960 s, respectively. To avoid variances in the photobleaching rate, we adjusted the laser power to 1.13 mW before starting each measurement.

To track fast-moving molecules and finally determine diffusion coefficients and bound fractions, we recorded movies with a short exposure time of 10 ms, with a total frame cycle time of 11.7 ms. To avoid high background, we activated the fluorophore with 0.05 mW UV illumination for a short time period of 1 ms between two 638 nm exposures. Since we only activated a small fraction of fluorophores in every frame, it was feasible to record 20,000 frames without decreasing density of activated molecules.

## Single-molecule data analysis

We used TrackIT to identify, localize, and track single molecules and perform diffusion analysis[39]. For time-lapse imaging movies, we used the threshold factor 4.5 to identify spots. For connecting bound molecules through consecutive frames, we set the tracking radius to 0.9 pixels (0.1 s tl), 1.19 pixels (0.4 s tl), 1.75 pixels (1.6 s tl), and 2.8 pixels (6.4 s tl). The minimum tracking length was 3 frames for 0.1 s tl and 0.4 s tl and 2 frames for the other tl conditions. We allowed 2 gap frames without detection for 0.1 s tl and 1 gap frame for longer tl conditions. The gap frame was only allowed if the track already included 2 frames before the gap.

For 11.7 ms continuous movies, we detected spots with a threshold factor of 4. We connected tracks using a tracking radius of 7, a minimum track length of 2, gap frame of 1, and track length before gap frame of 2. To determine diffusion coefficients and fractions, we fitted the cumulative survival time distribution with a three-exponential Brownian diffusion model[82]. The bin size of the distribution was set to 1120 which corresponded to 1 nm. For analysis we considered the first 5 jumps per track to avoid over-representation of immobile molecules and discarded jumps over gap frames. We estimated errors of diffusion coefficients $D_{1,2,3}$ and fractions $A_{1,2,3}$ by repeating diffusion analysis 500 times using 80% of randomly chosen jump distances. The amplitude $A_1$ of the slowest diffusion component represented the overall bound fraction $f_b$ of the tracked molecules. The unbound fraction of tracked molecules is then given by $p_f = 1 - f_b$.

## Analysis of survival time distributions using GRID

We inferred dissociation rate spectra of HT-RBPJ variants by analyzing survival time distributions from time-lapse imaging with GRID[40]. In brief, GRID reveals the amplitudes of $l$ dissociation rates $k_{off,l}$ associated with $l$ binding classes from fluorescence survival time distributions by an inverse Laplace transformation. Initially, GRID reveals the event spectrum, whose amplitudes $A^e_l$ represent the relative frequency of binding events that occur for a certain binding class $l$ over the observation period. $A^e_l$ is given by[40]

$$A^e_l = \frac{k^*_{on,l}}{\sum_l k^*_{on,l}} \qquad (1)$$

where $k^*_{on,l} = k_{on,l}[D_l]$, $k_{on,l}$ is the bimolecular association rate to a binding site of class $l$, and $[D_l]$ is the concentration of unoccupied binding sites $D_l$ of type $l$. By dividing the amplitudes with the corresponding rates and renormalization, the event spectrum is converted to the state spectrum, with amplitudes $A^s_l$[40]:

$$A^s_l = \frac{k^*_{on,l}/k_{off,l}}{\sum_l k^*_{on,l}/k_{off,l}} = \frac{[TD_l]}{\sum_l [TD_l]} \qquad (2)$$

where $[TD_l]$ is the concentration of transcription factors $T$ bound to binding sites $D_l$ of class $l$. The amplitudes $A^s_l$ of the state spectrum reflect the probability to find a molecule in a certain binding class $l$ at any time snapshot. Together with the overall bound fraction $f_b$, we obtain the overall fraction of molecules binding to a binding site of class $l$: $p_{b,l} = f_b * A^s_l$.

We estimated the error of dissociation rate spectra by repeating the GRID analysis 499 times with 80% of randomly chosen survival times for each GRID run.

For HT-RBPJ-WT, we obtained the search time $\tau_{search}$ from Eq. 7 by calculating $k^*_{on,u}$ and $N_{trial}$ with Eqs. 6 and 8 and inserting the measured parameters $A^e_s$, $k_{off,u}$, $p_f$, and $p_{b,u}$. The parameters entering the model and the resulting target site search time can be found in Supplementary Table 6.

## Calculation of the target site search time for the search mechanism of facilitated diffusion

In a commonly assumed model of transcription factor–DNA interactions, the transcription factor has two different classes of binding sites on DNA, unspecific and specific[1,83]. It thus performs the unspecific and specific binding reactions:

$$[T] + [D_u] \underset{k_{off,u}}{\overset{k_{on,u}}{\rightleftarrows}} [TD_u] \qquad (3)$$

$$[T] + [D_s] \underset{k_{off,s}}{\overset{k_{on,s}}{\rightleftarrows}} [TD_s] \qquad (4)$$

Where $[T]$ is the concentration of the unbound protein, $[D_u]$ and $[D_s]$ are the concentrations of unoccupied unspecific and specific binding sites, $[TD_u]$ and $[TD_s]$ are the concentrations of protein-bound binding sites, $k_{on,u}$ and $k_{on,s}$ are the bimolecular association rates and $k_{off,u}$ and $k_{off,s}$ are the dissociation rates (see analysis of survival time distributions).

The dissociation constant of either reaction is given by

$$K_{d,i} = \frac{k_{off,i}}{k_{on,i}} = \frac{[T][D_i]}{[TD_i]} \qquad (5)$$

with $i = u,s$. Thus, there is a relation between the ratio of kinetic rates and the ratio of the unbound fraction $p_f$ (see single-molecule data analysis) to the bound fraction $p_{b,i}$ (see analysis of survival time

distributions) of the DNA-binding protein:

$$\frac{k_{off,i}}{k_{on,i}[D_i]} = \frac{[T]}{[TD_i]} = \frac{p_f}{p_{b,i}} \tag{6}$$

We set $k^*_{on,i} = k_{on,i}[D_i]$ in the following.

To search for their specific target sites, transcription factors follow a search mechanism of facilitated diffusion including three-dimensional diffusion in the nucleoplasm and lower-dimensional exploration of the local environment[10,41]. Speed-up of the search process might occur via transient interactions including unspecific binding, sliding or hopping on DNA, association to cofactors, or nuclear structures such as protein clusters. We previously devised a three-state model of the target site search mechanism of facilitated diffusion[45] (Fig. 3a). Within this model, the transcription factor may occupy the three states: unbound (or free), unspecifically bound, and specifically bound. Unspecific binding with association rate $k^*_{on,u} = k_{on,u}[D_u]$ and dissociation rate $k_{off,u}$ may occur both at unspecific binding sites and in the vicinity of a specific target site. Specific binding may be achieved by either transition of the transcription factor from unspecific binding to specific binding with the microscopic on-rate $k_{u-s}$, or by direct association with the association rate $k^*_{on,s} = k_{on,s}[D_s]$. These two pathways are combined in the effective specific association rate $k^*_a$. Dissociation from a specific binding site may occur either by transition to the unspecifically bound state with the microscopic off-rate $k_{s-u}$, or by direct dissociation with dissociation rate $k_{off,s}$. These two pathways are combined in the effective specific dissociation rate $k_d = 1/\tau_s$. The target site search time $\tau_{search} = 1/k^*_a$ can be found as the inverse of the effective specific association rate. It is the average time a transcription factor needs to find any of the specific target sites and includes multiple rounds of unspecific binding and unbinding at unspecific binding sites or in the vicinity of the specific target site, before a final transition to specific binding occurs. While our model was originally devised to include binding to unspecific and specific DNA sequences, the transient unspecifically bound state may represent any rate-limiting transient interaction, be it with DNA, cofactors or other nuclear structures.

The association and dissociation rates of the three-state model are coupled by detailed balance[45]:

$$\frac{k^*_{on,u}}{k_{off,u}} \frac{k_{u-s}}{k_{s-u}} = \frac{k^*_{on,s}}{k_{off,s}} \tag{7}$$

Of the parameters entering the three-state model of facilitated diffusion, the unspecific dissociation rate $k_{off,u}$ and the effective specific dissociation rate $k_d$ are experimentally accessible. They are found by GRID analysis of the fluorescence survival time distributions of the transcription factor binding times measured with time-lapse microscopy (see analysis of survival time distributions). Moreover, the fractions of unbound ($p_f$), unspecifically bound ($p_u$) and specifically bound ($p_s$) molecules are experimentally accessible and found by diffusion measurements and GRID analysis (see analysis of single-molecule data and analysis of survival time distributions).

The unspecific association rate $k^*_{on,u}$ can be obtained by:

$$k^*_{on,u} = \frac{p_u}{p_f} k_{off,u} \tag{8}$$

To compare the search times of different HT-RBPJ variants despite we did not have access to the rates $k_{u-s}$, $k_{s-u}$, $k^*_{on,s}$, and $k_{off,s}$, we fixed the ratio $k_{off,s}/k_d$ for all variants to 0.0705, corresponding to a target site search time ten times faster than direct specific association. The direct

specific association rate $k^*_{on,s}$ can be obtained with Eq. 6:

$$k^*_{on,s} = \frac{p_s}{p_f} k_{off,s} \tag{9}$$

In the three-state model, the direct specific dissociation rate $k_{off,s}$ is coupled to $k^*_{on,u}$, $k_{u-s}$, and $k_d$ by[45]:

$$k_{off,s} = k_d \left(1 + \frac{k^*_{on,u}}{k^*_{on,s}} \frac{k_{u-s}}{k_{off,u}} \frac{k_{off,u} - k_d}{k_{u-s} + k_{off,u} + k_d}\right) \tag{10}$$

Thus, for given $k^*_{on,u}$, $k_d$, and $k_{off,s}$, the microscopic on-rate $k_{u-s}$ can be obtained:

$$k_{u-s} = \frac{(k_d - k_{off,u})(k_d/k_{off,s} - 1)k^*_{on,s}k_{off,u}/k^*_{on,u}}{(k_d - k_{off,u}) + (k_d/k_{off,s} - 1)k^*_{on,s}k_{off,u}/k^*_{on,u}} \tag{11}$$

and the microscopic off-rate $k_{s-u}$ is found in Eqs. 11 and 7 of detailed balance:

$$k_{s-u} = \frac{k^*_{on,u}}{k_{off,u}} \frac{k^*_{on,s}}{k_{off,s}} k_{u-s} \tag{12}$$

Finally, the target site search time $\tau_{search}$ of the three-state model is given by[45]:

$$\tau_{search} = \frac{1}{k^*_a} = \frac{k^*_{on,u} + k_{off,u} + k_{u-s}}{k^*_{on,s} k_{off,u} + \left(k^*_{on,s} + k^*_{on,u}\right) k_{u-s}} \tag{13}$$

The parameters entering the model and the resulting target site search times can be found in Supplementary Table 7.

To obtain $\tau_{search}$ for a given direct specific association rate $k^*_{on,s}$, we solved Eq. 9 for $k_{off,s}$ and followed the subsequent steps to calculate the target site search time.

**Construction of the binding energy landscape**

We determined the binding energy landscape of a transcription factor from the experimentally accessible kinetic rates $k_{off,u}$ and $k_d$, the measured unbound and bound fractions $p_f$, $p_u$, and $p_s$, and the kinetic rates $k^*_{on,u}$, $k_{u-s}$, $k_{s-u}$, and $k^*_{on,s}$ obtained from the three-state model of facilitated diffusion. Importantly, while binding energies are often compared at standard concentrations of binding sites, we judged it more relevant to compare the binding energies of transcription factor variants at the respective concentration of binding sites they actually have in the nucleus. The binding energy $\Delta G_u$ of the unspecifically bound state is given by:

$$\Delta G_u = -k_B T ln\left(\frac{k^*_{on,u}}{k_{off,u}}\right) = -k_B T ln\left(\frac{p_u}{p_f}\right) \tag{14}$$

with the Boltzmann constant $k_B$ and the unit of thermal energy $k_B T$. Analogously, the binding energy $\Delta G_s$ of the specifically bound state is given by:

$$\Delta G_s = -k_B T ln\left(\frac{k^*_{on,s}}{k_{off,s}}\right) = -k_B T ln\left(\frac{p_s}{p_f}\right) \tag{15}$$

To compare the relative height of energy barriers of the various association and dissociation transitions, we assumed a similar frequency factor $k_A$ for all transitions. The energy barriers of unspecific binding ($\Delta G_{f \to u}$), of switching from unspecific to specific binding

$(\Delta G_{u\to s})$, and of specific binding $(\Delta G_{f\to s})$ are then given by:

$$\Delta G_{f\to u} = -k_B T ln\left(\frac{k^*_{on,u}}{k_A}\right) \quad (16)$$

$$\Delta G_{u\to s} = -k_B T ln\left(\frac{k_{u-s}}{k_A}\right) + \Delta G_{f\to u} \quad (17)$$

$$\Delta G_{f\to s} = -k_B T ln\left(\frac{k^*_{on,s}}{k_A}\right) \quad (18)$$

## Reporting summary
Further information on research design is available in the Nature Portfolio Reporting Summary linked to this article.

## Data availability
ChIP-Seq data was uploaded to the Gene Expression Omnibus repository (GEO accession number: GSE249973). Single-particle tracking data are freely available at Data Dryad repository at https://doi.org/10.5061/dryad.mkkwh716k[84]. Data supporting the findings of this manuscript are additionally available from the corresponding authors upon request. Source data are provided as a Source Data file. Source data are provided with this paper.

## Code availability
The single-molecule tracking software TrackIt is available on GitLab. (https://gitlab.com/GebhardtLab/TrackIt) and Zenodo (https://zenodo.org/records/7092296)[85].

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

## Acknowledgements

We thank Jutta Hegler for help with Lentivirus production and Sabine Schirmer for excellent technical assistance, Devin Assenheimer for help with error calculations and Jonas Coßmann and Tobias Bischof for supportive discussions (all Ulm University). Halo-SiR ligand was kindly provided by K. Johnsson (Max Planck Institute for Medical Research). pLV-tetO-Oct4 was a gift from Konrad Hochedlinger (Massachusetts General Hospital), pMD2.G and psPAX2 were a gift from Didier Trono (Ecole Polytechnique Fédérale de Lausanne). pSpCas9(BB)-2A-Puro (PX459) V2.0 was a gift from Feng Zhang. We thank the Core Facility FACS of Ulm University for their help with cell sorting, with special thanks to Dr. Simona Ursu, Dr. Sarah Warth, and Daniela Froelich. The work was funded by the Deutsche Forschungsgemeinschaft (DFG, German Research Foundation no. 427512076 to J.C.M.G. and F.O., no. 468578170 and 422780363 SPP 2202 to J.C.M.G.), the European Research Council (ERC) under the European Union's Horizon 2020 Research and Innovation Program (no. 637987 ChromArch to J.C.M.G.), and the German Cancer Aid (#70114289 to F.O.) R.A.K. acknowledges funding by NSF/MCBBSF grant #1715822, T.B. acknowledges funding by the DFG TRR81- A12 (#109546710), a research grant of the University Medical Center Gießen and Marburg (UKGM), the LOEWE research cluster iCANx, and the Excellence Cluster for Cardio Pulmonary System (ECCPS) in Gießen. M.B. acknowledges funding by the Forschungscampus Mittelhessen. Support by the Collaborative Research Centres 1074 (DFG no 217328187), 1279 (DFG no. 316249678) and 1506 (DFG no. 450627322) and the Center for Translational Imaging MoMAN of Ulm University (DFG no. 447235146) is acknowledged.

## Author contributions

F.O. and J.C.M.G. conceived the project; D.H., P.H., F.O. and J.C.M.G. designed the project; F.F and B.D.G. generated the knockout cell lines; P.H. and F.O. performed luciferase assays; F.F., B.D.G. and T.B. performed ChIP-Seq experiments; T.F., B.D.G. and M.B. analyzed ChIP-Seq data; R.A.K. analyzed the RBPJ [Su(H)] structure; D.H. performed the single-molecule measurements; D.H. and J.C.M.G. analyzed the single-molecule data with contributions from K.Z.; D.H., P.H., F.O. and J.C.M.G wrote the manuscript with comments from all authors.

## Funding

## Competing interests

The authors declare no competing interests.
