## [Transparent Peer Review file · Nature Communications]

Effective in vivo binding energy landscape illustrates kinetic stability of RBPJ-DNA binding

Corresponding Author: Professor J. Christof Gebhardt

Version 0:

Reviewer comments:

Reviewer #1

(Remarks to the Author)

The authors present a study on the targeting kinetics of the transcription factor RBPJ to specific DNA sequences, utilizing various mutations in the DNA-binding domain and the cofactor binding interface. By employing techniques such as ChIP, molecular biology methods, and single-particle tracking (SPT), they achieve highly sensitive detection of association and dissociation kinetics. Overall, the authors provide a thorough investigation of RBPJ's target search and binding kinetics. However, several points require further clarification:

- 1) There is an inconsistency in the results regarding the mutant F261A/L388A (FL/AA). Figure 1c shows more than 50% transcription activation, and Figures 2f and 2g indicate a similar specific residence time compared to WT. However, ChIP data in Figures 1d and 1e show a significant reduction in binding (less than 30% of WT). My understanding is that the fraction of binding with specific (longer) residence time should correlate with ChIP results, yet Figures 1b, 1c, and 2f do not align with the ChIP data in Figures 1d and 1e. The authors should address this discrepancy.
- 2) In Supplemental Figure 2a, the input (before IP) and output (after IP) data should be presented in separate columns with two rows for each antibody (anti-FLAG and anti-RBPJ). The current presentation, with input and output data in the same column, is unconventional and confusing.
- 3) It is unclear whether FLAG-NICD overexpression increases the level of NICD binding with RBPJ. Since endogenous NICD is still present, understanding how FLAG-NICD overexpression enhances the binding of RBPJ with NICD is crucial. The ideal approach would be to immunoprecipitate HT-RBPJ and detect NICD (preferably using an antibody against endogenous NICD to assess total binding) via western blotting. Additionally, the FL/AA and RFL/HAA mutants are expected to show reduced binding with NICD, and the authors should present this data concurrently.
- 4) In Supplemental Figure 3b, the expression level of RBPJ in wild-type cells, RBPJ KO cells, and RBPJ KO cells with mutant expression should be shown. This is necessary to confirm that the level of HT-RBPJ expression is comparable to the wild-type level, as determined by western blotting.
- 5) In line 189, the authors state, "This highlights the importance of the DNA residence time for the functioning of TFs." The causal relationship between DNA binding and transcriptional activity remains unclear. While reduced DNA binding can lower transcriptional activity, it is also possible that a loss of transcriptional activation ability may decrease residence time.
- 6) In lines 246-247, the authors state, "Thus, in addition to the DNA residence time, the target site search time is important for RBPJ function." A more specific conclusion would be beneficial. For example, an increased target search time may be due to a reduced ability for stable binding, leading to unsuccessful transcriptional activation.
- 7) A deeper discussion on how kinetic, rather than thermodynamic, binding stability contributes to transcription is needed. For instance, does kinetic binding better regulate dynamic transcriptional responses to signaling? What are the advantages of kinetic over thermodynamic stability in TF activation?

Minor concerns:

- 8) Lines 62-64: The term "the most important DNA binding hub" seems subjective. It should be rephrased to a more objective term such as "core binding hub."
- 9) Lines 165-167: The sentence "We therefore reasoned..... either unspecific or specific binding sites" is unclear and needs rephrasing for clarity.

Reviewer #2

(Remarks to the Author)

In this manuscript, Huynh and Hoffmeister et al tackle important questions in the transcription factor (TF) field by using the RBPJ-Notch signaling system as an experimental model. By doing single-molecule tracking (SMT) in live cells of different RBPJ mutants, in combination with sophisticated data analysis pipelines, they conclude that cofactor binding is an important parameter in DNA binding specificity. Interestingly, they propose that in vivo DNA TF binding follows a kinetic rather than a thermodynamic model, providing new insights into TF-chromatin interactions. The manuscript is well-written and should be highly interesting to the gene regulation community. Below are some major and minor comments that I hope the authors can properly address to improve the manuscript.

Major comments

1. If I understand correctly, all SMT experiments involving RAPBJ were performed in the absence of a signal that activates the Notch-signaling pathway. I am assuming this is the case as I couldn't find any details in methods of whether NICD was transfected in SMT experiments. In other words, all the dynamic information for RAPBJ was obtained when the TF was in its "repressor mode", or in some cases in the absence of the corepressor (SHARP knockouts). Are the authors assuming that RAPBJ dynamics will remain the same when becoming active? Have the authors compared the SMT behavior of RAPBJ in the presence and absence of NICD? This might constitute an important control for the authors to consider doing.
2. If the SMT experiments were indeed performed in the absence of Notch activation, the correlation between the transcriptional activity of the active RAPBJ (co-transfected with NICD) vs the specific residence time of the TF in its inactive state (Fig. 2h) raises some experimental design concerns. Shouldn't the correlation be tested when the TF is transcriptionally active?
3. May I ask for the authors to expand here on how they discriminate between specific and non-specific binding to plot Fig. 2e-h? According to lines 165-171, it seems it is assumed that specific binding is longer than non-specific binding, and then the KRS/EHD DNA-binding mutant to establish a threshold for what would be considered specific and non-specific. There are a few studies and review articles that claim that specific and non-specific binding cannot be discriminated based on residence times (PMID: 26151127, PMID: 26387491, PMID: 31283382, PMID: 33592625, PMID: 37540844). Perhaps the discussion section could tackle this apparent controversy.
4. Fig. 2f-g. The authors perform experiments in the SHARP-less cells that have endogenous RAPBJ. Could heterodimers between endogenous RAPBJ and mutant RAPBJ interfere with the observed outcome?
5. The authors conclude (lines 358-363) that cofactors might play a role in guiding RBPJ to its cognate sites, based on the SMT phenotype of the FL/AA mutant. I wonder if the authors have tried to rescue this phenotype by using their RAPBJ-VP16 chimera. In short, test whether the FL/AA mutant behaves similarly to WT-VP16 if you add the VP16 coactivator domain to the mutant. In this way, you can rule out confounding effects of the mutation that might affect the protein beyond the inability to interact with cofactors/corepressors. This is just a suggestion and not a mandatory request experiment.
6. lines 193-196. The authors claim that in the nucleus TF searches through 3D diffusion in the nucleoplasm and 1D dimensional sliding along unspecific DNA. The reference is purely theoretical (ref.39). I would argue that there is no direct evidence yet that eucaryotic TFs perform 1D sliding on chromatin inside the nucleus. Please provide a reference if I am mistaken. Therefore, I suggest you rephrase this statement more carefully, clarifying this mechanism has been hypothesized but not yet proven.
7. I think it will help the readers to include in the discussion a paragraph pointing to the limitations of the study and summarizing the assumptions of their theoretical model.

Minor Comments:

1. Fig. 1f-g. Ven diagrams are not at scale and therefore can be misleading at first glance
2. Figure S5d-f: Why do the ChIP peaks look so "spread"? Usually, ChIP-peaks are narrower (i.e. sharper). Is this something particular to this TF?
3. Fig. S5f: missing the gene name in browser shot (left gene)
4. Fig. 2d: blue line is too thin to be noticeable. Please make it darker and thicker
5. S2 movie. I've noticed quite a few particles that appear to enter and exit the nucleus as if there were no barrier at all. This is quite odd behavior. Could the authors please comment as to whether this is common in the entire dataset? Is it possible the nuclear region has not been properly drawn/established?

Version 1:

Reviewer comments:

Reviewer #1

(Remarks to the Author)

The authors have largely addressed the issues raised; however, a couple of concerns remain. While the authors provided WB data showing equivalent FLAG-NICD/SHARP expression levels using a FLAG antibody in the input (before IP) in Supplementary Figure 2b/e, the corresponding FLAG-NICD data for the output (after IP) is missing. As the ratio of band intensities between the FLAG antibody and RBPJ in the output reflects the binding affinity, it is essential to include WB data using the FLAG antibody in the output (after IP) in Supplementary Figure 2a/d.

Additionally, the authors reported 1.4- and 1.8-fold overexpression of HT-RBPJ variants in RBPJ KO cells compared to endogenous RBPJ in wild-type cells. I am concerned that this higher expression of HT-RBPJ variants could result in inaccurate bound fraction values.

Reviewer #2

(Remarks to the Author)

The authors have answered all my concerns.

I have no further comments

Response to Reviewers

We thank the reviewers for their helpful and constructive criticism. In the following, we address each point individually. We highlighted the corresponding changes in the manuscript.

Reviewer #1 (Remarks to the Author):

The authors present a study on the targeting kinetics of the transcription factor RBPJ to specific DNA sequences, utilizing various mutations in the DNA-binding domain and the cofactor binding interface. By employing techniques such as ChIP, molecular biology methods, and single-particle tracking (SPT), they achieve highly sensitive detection of association and dissociation kinetics. Overall, the authors provide a thorough investigation of RBPJ's target search and binding kinetics. However, several points require further clarification:

Thank you for your appreciation and the fair assessment of our work.

1) There is an inconsistency in the results regarding the mutant F261A/L388A (FL/AA). Figure 1c shows more than 50% transcription activation, and Figures 2f and 2g indicate a similar specific residence time compared to WT. However, ChIP data in Figures 1d and 1e show a significant reduction in binding (less than 30% of WT). My understanding is that the fraction of binding with specific (longer) residence time should correlate with ChIP results, yet Figures 1b, 1c, and 2f do not align with the ChIP data in Figures 1d and 1e. The authors should address this discrepancy.

We thank the reviewer for pointing out the discrepancy between the ChIP-Seq data and activity/SMT data. Indeed, since the DBD of FL/AA is undisturbed, similar binding properties as for wt are expected. While we measured comparable residence times and only slightly reduced bound fractions using SMT, ChIP-Seq showed much lower binding. We carefully reanalyzed our data, but could not find any error. However, we noticed that the cofactor mutants (FL/AA and RFL/HAA) showed cytoplasmic localization in addition to the expected nuclear localization that we observed for all other variants. Lower nuclear levels of FL/AA and RFL/HAA can lead to less ChIP-Seq signal. In contrast, SMT residence times and bound fractions would be unaffected, as they are obtained independent of or relative to the nuclear pool, respectively.

Given this discrepancy, we now toned down our conclusion in the abstract: „Moreover, our data point to the possibility that cofactor binding, in addition to DNA binding, contributes to target site specificity“, and in the main text. Moreover, we now mention the potential reason for the discrepancy in the main text.

2) In Supplemental Figure 2a, the input (before IP) and output (after IP) data should be presented in separate columns with two rows for each antibody (anti-FLAG and anti-RBPJ). The current presentation, with input and output data in the same column, is unconventional and confusing.

We apologize for the confusing representation. As suggested, we now present the expression control separately.

3) It is unclear whether FLAG-NICD overexpression increases the level of NICD binding with RBPJ. Since endogenous NICD is still present, understanding how FLAG-NICD overexpression enhances the binding of RBPJ with NICD is crucial. The ideal approach would be to immunoprecipitate HT-RBPJ and detect NICD (preferably using an antibody against endogenous NICD to assess total binding) via western blotting. Additionally, the FL/AA and RFL/HAA mutants are expected to show reduced binding with NICD, and the authors should present this data concurrently.

We agree that we should more directly control for reduced binding of FL/AA and RFL/HAA to HT-RBPJ, in addition to the activity assays, which already showed reduced activation. Given that unstimulated HeLa cells present a Notch-OFF status and therefore, endogenous active NICD is mostly absent in HeLa cells (PMID: 29440432), we preferred to pull down ectopic Flag-NICD to control cofactor binding capacities of RBPJ variants. As expected, WT and DBD mutant KRS showed much stronger signal than cofactor mutants FL/AA and RFL/HAA. In addition, we repeated the experiment with Flag-SHARP, and again observed reduced binding to cofactor mutants. Importantly, the results also show that HaloTag does not prevent RBPJ WT interaction with cofactors.

We included these controls in Supplementary Fig 3.

4) In Supplemental Figure 3b, the expression level of RBPJ in wild-type cells, RBPJ KO cells, and RBPJ KO cells with mutant expression should be shown. This is necessary to confirm that the level of HT-RBPJ expression is comparable to the wild-type level, as determined by western blotting.

As suggested, we now quantified overexpression of HT-RBPJ variants in RBPJ KO cells relative to endogenous RBPJ in wild type cells. We found values between 1.4 and 1.8-fold overexpression.

We included the quantification in the main text and as Supplementary Fig 3.

5) In line 189, the authors state, "This highlights the importance of the DNA residence time for the functioning of TFs." The causal relationship between DNA binding and transcriptional activity remains unclear. While reduced DNA binding can lower transcriptional activity, it is also possible that a loss of transcriptional activation ability may decrease residence time.

We agree with the reviewer that our experiments do not show causality. We therefore changed this sentence to one that works bidirectional: "This suggests that the DNA residence time of a TF is an indicator for its functionality."

6) In lines 246-247, the authors state, "Thus, in addition to the DNA residence time, the target site search time is important for RBPJ function." A more specific conclusion would be beneficial. For example, an increased target search time may be due to a reduced ability for stable binding, leading to unsuccessful transcriptional activation.

The reviewer is right that the search time depends on the duration of unspecific binding, however, it does not depend on the duration of specific binding (Ref Hettich JTB 2018). In addition, the relation to unspecific binding is not monotonous. Therefore, a more specific conclusion cannot be made. Nevertheless, to adapt to comment 5 of the reviewer, we now refined our conclusion: "Thus, in addition to the DNA residence time, the target site search time is an indicator of RBPJ function. Together, both parameters determine the occupancy of the TF binding sites."

7) A deeper discussion on how kinetic, rather than thermodynamic, binding stability contributes to transcription is needed. For instance, does kinetic binding better regulate dynamic transcriptional responses to signaling? What are the advantages of kinetic over thermodynamic stability in TF activation?

As suggested, we now enhanced the discussion of our findings. Of note, kinetic and thermodynamic stability refer to different regimes of the binding equilibrium, they do not denote differences in TF function: "While the residence time is mostly hard-coded in the structure of the TF, the target site search time can be modulated by altering the concentration of TFs. Thus, by expressing more TFs in response to internal or external stimuli, the cell is able to decrease the TF search time and thereby shift the binding equilibrium towards more favorable binding. In case the search time becomes shorter than the residence time, binding becomes thermodynamically stable and bound fractions can increase above 0.5. The TF concentrations found in the cell presumably reflect the need to balance sufficiently high TF binding with minimal TF concentrations to economize limited resources."

Minor concerns:

8) Lines 62-64: The term "the most important DNA binding hub" seems subjective. It should be rephrased to a more objective term such as "core binding hub."

As suggested, we changed the expression to: "..., which functions as the core DNA binding hub."

9) Lines 165-167: The sentence "We therefore reasoned..... either unspecific or specific binding sites" is unclear and needs rephrasing for clarity.

We changed the sentence and hope it is now clearer: "We therefore reasoned that the longest residence time of HT-RBPJ-KRS/EHD and the corresponding dissociation rate cluster might mark the border between unspecific and specific binding regimes. We thus used the dissociation spectrum of HT-RBPJ-KRS/EHD to sort dissociation of HT-RBPJ variants into dissociation from either unspecific or specific binding sites (Figure 2d)."

Reviewer #2 (Remarks to the Author):

In this manuscript, Huynh and Hoffmeister et al tackle important questions in the transcription factor (TF) field by using the RBPJ -Notch signaling system as an experimental model. By doing single-molecule tracking (SMT) in live cells of different RBPJ mutants, in combination with sophisticated data analysis pipelines, they conclude that cofactor binding is an important parameter in DNA binding specificity. Interestingly, they propose that in vivo DNA TF binding follows a kinetic rather than a thermodynamic model, providing new insights into TF-chromatin interactions. The manuscript is well-written and should be highly interesting to the gene regulation community. Below are some major and minor comments that I hope the authors can properly address to improve the manuscript.

We thank the reviewer for the positive assessment of our manuscript.

Major comments

1. If I understand correctly, all SMT experiments involving RAPBJ were performed in the absence of a signal that activates the Notch-signaling pathway. I am assuming this is the case as I couldn't find any details in methods of whether NICD was transfected in SMT experiments. In other words, all the dynamic information for RAPBJ was obtained when the TF was in its "repressor mode", or in some cases in the absence of the corepressor (SHARP knockouts). Are the authors assuming that RAPBJ dynamics will remain the same when becoming active? Have the authors compared the SMT behavior of RAPBJ in the presence and absence of NICD? This might constitute an important control for the authors to consider doing.

The reviewer is right in that the residence times and search times of RBPJ variants were determined in absence of NICD. Since the structure of the RBPJ-DNA binding interface is similar when SHARP or NICD are bound (PMID: 30673607), and none of the known cofactors of RBPJ that bind to the BTB-CTD interface (e.g. NICD or SHARP) show potentially stabilizing binding to DNA, we do not expect that the residence times of RBPJ variants or their relative order show larger changes if NICD is absent or present. Measuring the residence times in presence of NICD is however out of the scope of the current manuscript.

2. If the SMT experiments were indeed performed in the absence of Notch activation, the correlation between the transcriptional activity of the active RAPBJ (co-transfected with NICD) vs the specific residence time of the TF in its inactive state (Fig. 2h) raises some experimental design concerns. Shouldn't the correlation be tested when the TF is transcriptionally active?

As stated above, when we correlated the residence times with transcriptional activity in presence of NICD, we assumed that the dynamic parameters did not change. However, we agree that showing this correlation can be misleading. We therefore now show the activity of RBPJ-VP16 fusions versus residence time. Importantly, such fusions are a common approach to test the functionality of binding of repressing TFs (PMID: 10954611, PMID: 25150167) and therefore allowed us to characterize the functional binding of RBPJ variants in absence of NICD. We edited the text and figures accordingly.

3. May I ask for the authors to expand here on how they discriminate between specific and non-specific binding to plot Fig. 2e-h? According to lines 165-171, it seems it is assumed that specific binding is longer than non-specific binding, and then the KRS/EHD DNA-binding mutant to establish a threshold for what would be considered specific and non-specific. There are a few studies and review articles that claim that specific and non-specific binding cannot be discriminated based on residence times (PMID: 26151127, PMID: 26387491, PMID: 31283382, PMID: 33592625, PMID: 37540844). Perhaps the discussion section could tackle this apparent controversy.

In the mentioned publications, the survival time distributions of several TFs, endogenous and bacterial, were measured in mammalian cells. Similar to the measurements reported in these publications, we observed survival time distributions of RBPJ-WT and mutant variants that stretch over several orders of magnitude. In particular, also mutants with disturbed DNA binding domain showed long binding events. In this respect, there is no discrepancy between our and the previously published measurements. The difference is mainly in the way of analysis. While the mentioned manuscripts used a power law to fit survival time distributions, we chose to use the GRID method, as this method yields dissociation rates which can be further used in kinetic modeling. This is not possible with the power law model.

Importantly, our discrimination of specific and unspecific binding is initially based on the functional differences between RBPJ-WT and the mutants with disturbed DNA binding domain, as well as the differential binding observed in ChIP experiments. We observed that the mutant KRS did not any more support transcriptional activity with a fused VP16 domain. It also showed strongly reduced binding to RBPJ sites in ChIP without de novo binding. Next, we observed that very long binding events present with RBPJ WT were missing in the mutant KRS. Comparing this difference in residence times between KRS and RBPJ-WT with the ChIP and VP16 experiments allowed us to conclude that specific binding is associated with the longest residence times. This might be different for the previously published TFs, for which a functional control with mutants has not been made. Currently, we cannot exclude that short binding events of RBPJ-WT also have functional consequences.

As suggested, we now more clearly articulated our arguments to discriminate between specific and unspecific binding, and expanded the discussion to include this topic.

4. Fig. 2f-g. The authors perform experiments in the SHARP-less cells that have endogenous RAPBJ. Could heterodimers between endogenous RAPBJ and mutant RAPBJ interfere with the observed outcome?

There is no evidence in the literature that RBPJ forms homodimers or is active as dimers. If RBPJ formed dimers, homodimer formation of HT-RBPJ variants could still occur in cell lines lacking endogenous RBPJ. Therefore, we assume that the presence of endogenous RBPJ does not interfere with our measurements of HT-RBPJ - DNA binding times.

5. The authors conclude (lines 358-363) that cofactors might play a role in guiding RBPJ to its cognate sites, based on the SMT phenotype of the FL/AA mutant. I wonder if the authors have

tried to rescue this phenotype by using their RAPBJ-VP16 chimera. In short, test whether the FL/AA mutant behaves similarly to WT-VP16 if you add the VP16 coactivator domain to the mutant. In this way, you can rule out confounding effects of the mutation that might affect the protein beyond the inability to interact with cofactors/corepressors. This is just a suggestion and not a mandatory request experiment.

We used the VP16 fusions to obtain a functional readout of RBPJ-DNA binding. The activation domain VP16 recruits the transcriptional machinery if the fusion protein is bound. Nevertheless, cofactors might still be able to bind to this fusion protein. Thus, we do not expect that the cofactor binding-impaired mutant FL/AA would behave similar to the RBPJ-WT if fused to VP16. We therefore restrained from performing the suggested experiment.

6. lines 193-196. The authors claim that in the nucleus TF searches through 3D diffusion in the nucleoplasm and 1D dimensional sliding along unspecific DNA. The reference is purely theoretical (ref.39). I would argue that there is no direct evidence yet that eucaryotic TFs perform 1D sliding on chromatin inside the nucleus. Please provide a reference if I am mistaken. Therefore, I suggest you rephrase this statement more carefully, clarifying this mechanism has been hypothesized but not yet proven.

We agree with the reviewer that the search mechanism of TFs is not well understood in mammalian cells. Nevertheless, as in procaryotes, direct association of a TF to a specific target site in the mammalian nucleus is slow and association is rather thought to proceed via a faster search mechanism combining three-dimensional diffusion in the nucleoplasm and lower-dimensional exploration of the local environment (PMID: 32413318, PMID: 34481381). Speed-up of the search process might occur via transient interactions including unspecific binding, sliding or hopping on DNA, association to cofactors, or nuclear structures such as protein clusters. While our theoretical model was originally devised to include binding to unspecific and specific DNA sequences, the transient unspecifically bound state may represent any rate-limiting transient interaction, be it with DNA, cofactors or other nuclear structures.

We now changed the corresponding text passages in the manuscript and methods section. In addition, we now clearly mention that the search mechanism of RBPJ is currently unknown, but that we assume that it follows a search mechanism described above.

7. I think it will help the readers to include in the discussion a paragraph pointing to the limitations of the study and summarizing the assumptions of their theoretical model.

As suggested, we now enhanced the discussion by sections on the partition into unspecific and specific binding and the model we employed to quantify the search time.

Minor Comments:

1. Fig. 1f-g. Ven diagrams are not at scale and therefore can be misleading at first glance

We agree that a scaled Venn diagram would be beneficial. However, because the number of binding sites varies by several orders of magnitude, such a diagram would be very difficult for the

reader to interpret. We now complemented the Venn diagrams with an UpSet plot in the supplementary Figure 2. An UpSet plot displays all possible combinations in a scaled format, making it well-suited for the quantitative analysis of data with multiple sets and large variability.

2. Figure S5d-f: Why do the ChIP peaks look so “spread”? Usually, ChIP-peaks are narrower (i.e. sharper). Is this something particular to this TF?

Thanks for pointing out the shape of the ChIP-seq peaks. The “spread out” appearance results from the window size (+/- 3kb) around the peak center. A smaller window makes the peaks appear more spread out, while a larger window narrows the peak. However, this is merely an optical effect and does not impact the biological interpretation. We prefer to keep this window size, as it is frequently used in other publications.

3. Fig. S5f: missing the gene name in browser shot (left gene)

We now added the gene name, Hey2.

4. Fig. 2d: blue line is too thin to be noticeable. Please make it darker and thicker

We now adjusted the thickness of the blue line.

5. S2 movie. I’ve noticed quite a few particles that appear to enter and exit the nucleus as if there were no barrier at all. This is quite odd behavior. Could the authors please comment as to whether this is common in the entire dataset? Is it possible the nuclear region has not been properly drawn/established?

When analyzing nuclear events, we try to be conservative in drawing the nuclear outline, to make sure no events outside of the nucleus are detected and blur the results. Thus, some molecules might leave the ROI, even if they are still nuclear. We now added a note in the methods section.

Response to Reviewers

Reviewer #1 (Remarks to the Author):

The authors have largely addressed the issues raised; however, a couple of concerns remain. While the authors provided WB data showing equivalent FLAG-NICD/SHARP expression levels using a FLAG antibody in the input (before IP) in Supplementary Figure 2b/e, the corresponding FLAG-NICD data for the output (after IP) is missing. As the ratio of band intensities between the FLAG antibody and RBPJ in the output reflects the binding affinity, it is essential to include WB data using the FLAG antibody in the output (after IP) in Supplementary Figure 2a/d.

As suggested, we now also show the Flag-NICD and FLAG-SHARP output and quantified the corresponding band intensities as suggested. Since affinities cannot be determined from Western blots, we now plot corresponding relative coimmunoprecipitation efficiencies.

We updated our Supplementary Figure 2 and the Methods section accordingly.

Additionally, the authors reported 1.4- and 1.8-fold overexpression of HT-RBPJ variants in RBPJ KO cells compared to endogenous RBPJ in wild-type cells. I am concerned that this higher expression of HT-RBPJ variants could result in inaccurate bound fraction values.

We already have addressed a potential influence of overexpression on the bound fraction in our original manuscript, and showed a corresponding control.

Nevertheless, as suggested, we now highlight such a possible artifact and the control showing no such effect even more:

“Overexpression of RBPJ variants compared to endogenous RBPJ might lead to saturation of binding sites, thereby artificially decreasing the observed bound fraction. However, a linear relation between bound molecules and all detected molecules confirmed that binding was not saturated in our overexpression conditions (Supplementary Figure 7f). Thus, the bound fraction should largely be unaffected at our levels of overexpression.”

Moreover, we added a similar discussion to a potential influence of overexpression on the number of identified binding sites in ChIP experiments:

“Overexpression of RBPJ variants compared to endogenous RBPJ might artificially increase the number of identified binding sites. Thus, the values we obtained represent an upper limit. However, the number of binding sites identified in the endogenous RBPJ control (17806) was larger than the one in the overexpressed HT-RBPJ reconstitution (15111), indicating that the number of identified binding sites was not overestimated at our level of overexpression.”

Reviewer #2 (Remarks to the Author):

The authors have answered all my concerns.
I have no further comments

We thank the reviewer for their approval of our work.